# EZH2/hSULF1 axis mediates receptor tyrosine kinase signaling to shape cartilage tumor progression

Zong-Shin Lin[1], Chiao-Chen Chung[2], Yu-Chia Liu[2], Chu-Han Chang[2], Hui-Chia Liu[2], Yung-Yi Liang[1], Teng-Le Huang[3], Tsung-Ming Chen[4], Che-Hsin Lee[5], Chih-Hsin Tang[1], Mien-Chie Hung[1,2,6]*, Ya-Huey Chen[1,2]*

[1]Graduate Institute of Biomedical Sciences, College of Medicine, China Medical University, Taichung, Taiwan; [2]Center for Molecular Medicine, China Medical University Hospital, Taichung, Taiwan; [3]Department of Biomedical Imaging and Radiological Science, College of Medicine, China Medical University, Taichung, Taiwan; [4]Department and Graduate Institute of Aquaculture, National Kaohsiung Marine University, Kaohsiung, Taiwan; [5]Department of Biological Sciences, National Sun Yat-sen University, Kaohsiung, Taiwan; [6]Department of Biotechnology, Asia University, Taichung, Taiwan

*For correspondence: mhung@cmu.edu.tw (M-ChieH); yahuey@mail.cmu.edu.tw (Y-HueyC)

Competing interest: The authors declare that no competing interests exist.

**Abstract** Chondrosarcomas are primary cancers of cartilaginous tissue and capable of alteration to highly aggressive, metastatic, and treatment-refractory states, leading to a poor prognosis with a five-year survival rate at 11 months for dedifferentiated subtype. At present, the surgical resection of chondrosarcoma is the only effective treatment, and no other treatment options including targeted therapies, conventional chemotherapies, or immunotherapies are available for these patients. Here, we identify a signal pathway way involving EZH2/SULF1/cMET axis that contributes to malignancy of chondrosarcoma and provides a potential therapeutic option for the disease. A non-biased chromatin immunoprecipitation sequence, cDNA microarray analysis, and validation of chondrosarcoma cell lines identified sulfatase 1 (*SULF1*) as the top EZH2-targeted gene to regulate chondrosarcoma progression. Overexpressed EZH2 resulted in downregulation of SULF1 in chondrosarcoma cell lines, which in turn activated cMET pathway. Pharmaceutical inhibition of cMET or genetically silenced cMET pathway significantly retards the chondrosarcoma growth and extends mice survival. The regulation of EZH2/SULF1/cMET axis were further validated in patient samples with chondrosarcoma. The results not only established a signal pathway promoting malignancy of chondrosarcoma but also provided a therapeutic potential for further development of effective target therapy to treat chondrosarcoma.

## Editor's evaluation

This fundamental work by Hung et al. substantially advances our understanding of the biology of chondrosarcoma, a primary cancer of cartilaginous tissue that can progress to highly aggressive, metastatic, and treatment-refractory tumor. The authors provided compelling to exceptional evidence to support the conclusion that EZH2/ hSULF1 axis mediates receptor tyrosine kinase signaling to shape cartilage tumor progression. This work will be of broad interest to cancer biologists and oncologists.

## Introduction

Chondrosarcoma, a malignant cartilaginous tumor, is the most common primary skeletal malignancy affecting bone (***Editorial and Consensus Conference Working Group, 2002***). Accumulation of mutations in molecules regulating cell growth control, programmed cell death, and DNA instability is

required for transformation from cartilage neoplasia to malignancy tumors. They are further classified into primary central and secondary peripheral chondrosarcomas based on their location in the bone (*Gelderblom et al., 2008*). The most common sites are bones of the pelvis, followed by the diaphysis or metaphysis of the proximal femur and humerus, distal femur, and ribs. The histologies in central and peripheral chondrosarcomas are similar, and all three different grades are discerned. They are the best predictor of clinical behavior at present (*Evans et al., 1977*). Metastasis is rare in Grade I chondrosarcomas, while metastases occur in 10% of Grade II and 71% of Grade III chondrosarcomas (*Björnsson et al., 1998*). Conventional chondrosarcomas are extremely resistant to chemotherapy and radiotherapy, resulting in limited treatment options (*Choy et al., 2012*; *Gelderblom et al., 2008*). Depending on the grade of the chondrosarcoma, a therapeutic surgical approach remains the only practical treatment (*Lee et al., 1999*). After surgery, patients with poor-quality life and for those inoperable patients (70%), radiation therapy alone is an option with uncertainty outcome (*Fromm et al., 2018*).

Additionally, the relatively poor vascularity results in more difficult delivering drugs than to other vascularized cancers (*Bovée et al., 2010*; *Gelderblom et al., 2008*; *Lee et al., 1999*). Due to rarity and various subtypes of chondrosarcoma, lack of interest from pharmaceutical industry obstructs the drug development and conduct of clinical trials for these patients (*Miwa et al., 2019*). Until now, there are no effective treatments for patients with inoperable or metastatic disease (70%), and therefore, it is an un-met medical need to develop new treatment modalities.

IGF inhibitors have been used to treat chondrosarcomas by increasing the rate of apoptosis (*Ho et al., 2009*; *Olmos et al., 2010*). Small molecule tyrosine kinase inhibitor dasatinib is known to reduce cell viability in chondrosarcoma cell lines by targeting SRC (*Schrage et al., 2009*). Recently, several target therapies were applied in phase I or II clinical trials in chondrosarcoma, including isocitrate dehydrogenase (IDH) inhibitor, tyrosine kinase inhibitors targeting vascular endothelial growth factor receptor (VEGFR), platelet derived growth factor receptor (PDGFR), insulin-like growth factor receptor (IGF-1R; *Truong et al., 2022*). These clinical trials are encouraging, however, remain in the early stage, and the clinical outcomes are yet to observe.

Enhancer of Zeste 2 (EZH2) is the major component of the polycomb repressor complex 2 (PRC2; *Kuzmichev et al., 2002*). PRC2 has been reported to repress gene expression through the histone methyltransferase EZH2-catalyzed trimethylation of histone 3 lysine 27 (H3K27) to establish repressive epigenetic marker (*Cao et al., 2002*; *Cao and Zhang, 2004*; *Kuzmichev et al., 2002*). EZH2 is required to maintain DNA methylation and suppress specific genes such as tumor suppressor genes by direct interaction of EZH2 and DNA methyltransferases (DNMT1, DNMT3A, and DNMT3B; *Sauvageau and Sauvageau, 2010*). EZH2 possesses the catalytic ability to transfer a methyl group from S-adenosyl methionine to H3K27 in the PRC2 and represses the expression of target genes including a broad range of genes such as those in the regulation of cell cycle, DNA damage repair, cell fate and differentiation, senescence, apoptosis, and cancer (*Cao et al., 2002*). Overexpression of EZH2 has been observed in lymphomas, sarcomas, and cancers of the breast, colon, liver, lung, and prostate (*Chang and Hung, 2012*; *Han et al., 2020*; *Li et al., 2009*; *Sauvageau and Sauvageau, 2010*; *Yamaguchi and Hung, 2014*). It has been reported that EZH2 mediates different types of human cancer through transcriptional, post-transcriptional, and post-translational stages, demonstrating that transcriptional factors, such as E2F, are able to bind to the promoters of EZH2 and (Embryonic ectoderm development)EED, which is necessary for E2F modulated-cell proliferation (*Bracken et al., 2003*; *Wei et al., 2011*).

EWS-FLI1, a fusion oncoprotein in Ewing's sarcoma, can transcriptionally activate EZH2 expression, and upregulated EZH2 expression plays a vital role in tumor growth and endothelial/neuroectodermal differentiation (*Richter et al., 2009*). More recently, *Taniguchi et al., 2012* demonstrated that EZH2 directly targets the Kruppel-like factor 2 (KLF2), which is a tumor suppressor, and interrupts its functions in cell cycle arrest and apoptosis, leading to tumorigenesis in the osteosarcoma mouse model (*Taniguchi et al., 2012*). In addition, a number of post-translational modifications also regulate EZH2-mediated signaling (*Cha et al., 2005*; *Chang et al., 2011*; *Nie et al., 2019*; *Wan et al., 2018*; *Wei et al., 2011*). These studies all point to a crucial role of EZH2 in tumorigenesis through a number of events in different human cancers including sarcoma. Notably, the dysregulation of differentiation programs of mesenchymal stem cells (MSCs) may lead to osteosarcomas and chondrosarcomas (*Mohseny and Hogendoorn, 2011*). Our previous results demonstrated that EZH2

might be a key modulator in bone development and sarcoma pathogenesis (*Chen et al., 2016*; *Wei et al., 2011*).

Human sulfatase 1 (SULF1) was characterized and revealed to desulfate cellular heparan sulfate proteoglycans (HSPGs). Sulfated HSPGs play a vital regulator of heparin binding growth factor signaling, e.g., FGF2, VEGF, HGF, PDGF, and heparin binding EGF (HB-EGF), also known as receptor tyrosine kinase (RTK) signaling (*Lai et al., 2003*). Therefore, enzymes that desulfate HSPGs could have a tumor suppressor effect via abolishment of ligand-receptor binding and downstream signaling. Complete loss or markedly attenuated expression of SULF1 appeared in cancer cell lines derived from breast, pancreas, kidney, and hepatocellular cancers, suggesting that downregulation of SULF1 is relatively common among epithelial cancers (*Lai et al., 2008*).

In this study, we integrated EZH2-chromatin immunoprecipitation (ChIP) sequence and cDNA microarray analysis then validated in multiple chondrosarcoma cell lines to identify *SULF1* as a top EZH2-targeted gene governing chondrosarcoma progression. We unravel the critical roles of SULF1 in suppressing tumor growth of chondrosarcoma by reducing RTK signaling pathway such as cMET which is known to increase malignancy and causes resistance to target therapy (*Chu et al., 2020*; *Du et al., 2016*; *Li et al., 2019*; *Sun et al., 2020*). Using pharmaceutical inhibitors targeting at EZH2 or cMET (*Yang et al., 2015*) to treat chondrosarcoma mice model significantly reduced tumor growth and prolonged mice survival. The results offer an opportunity for the development of effective target therapy to treat chondrosarcoma.

## Results

### EZH2 may have a role in chondrosarcoma development

Overexpression of EZH2 has been observed in sarcomas, lymphomas, and cancers of the breast, colon, liver, lung, and prostate (*Chang and Hung, 2012*; *Sauvageau and Sauvageau, 2010*). The abnormal differentiation programs of MSCs regulated by EZH2 results in osteosarcomas and chondrosarcomas (*Mohseny and Hogendoorn, 2011*). In addition, EZH2 was shown to govern bone development (*Chen et al., 2016*) and sarcoma pathogenesis (*Wei et al., 2011*). Thus, we hypothesize that EZH2 might have a role in the chondrosarcoma.

To examine the potential role of EZH2 in chondrosarcoma malignancy, we initially conducted western blot to analyze the expression of EZH2 in normal chondrocytes and chondrosarcoma cell lines. In contrast to normal chondrocytes, EZH2 was highly expressed in CH2879 and JJ012 chondrosarcoma cell lines as well as increased histone 3 trimethylation at lysine 27 (H3K27me3; *Figure 1A*). Due to lack of specific chondrosarcoma database of TCGA(The Cancer Genome Atlas), we further systematically examined the prognostic significance of EZH2 in sarcoma, a group including chondrosarcoma, using the median values for patient stratification analyzed by the Kaplan-Meier Plotter, a tool for meta-analysis-based biomarker assessment (*Figure 1B*). Consistently, the expression of *EZH2* correlated well with poor survival of sarcoma patients, and the justification of clinical relevance for EZH2 in chondrosarcoma by using tissue microarray was also shown later in Figure 8.

To validate the role of EZH2 in tumorigenicity of chondrosarcoma, we introduced sh*EZH2* or control shRNA by lentiviral system into chondrosarcoma cell lines (*Figure 1C and D*). Knockdown of *EZH2* reduced the proliferative, migrated, and colony-forming capacity of chondrosarcoma cell lines (*Figure 1E and I*) and supporting the notion that *EZH2* may have a role in chondrosarcoma development.

### *SULF1* is an EZH2-targeted gene and downregulated in chondrosarcoma

To identify EZH2-targeted genes in chondrosarcoma, we used cDNA microarray and EZH2-ChIP sequencing (ChIP-seq) profiling to identify the dysregulated genes which were targeted by EZH2 in chondrosarcoma cell lines. We first searched the genes that were both targeted by EZH2 (n=704) and associated with significant dysregulated genes from cDNA microarray in comparison with normal chondrocyte to JJ012 chondrosarcoma cell line (n=6771). Between these two pooled genes, 40 were overlapped. Because EZH2 is a gene silencer and catalyzes H3K27 trimethylation (H3K27me3) in the nucleus to modulate chromatin compaction followed by transcriptional suppression of downstream genes. Thus, we hypothesized that upregulated-*EZH2* may silence tumor-suppressive genes.

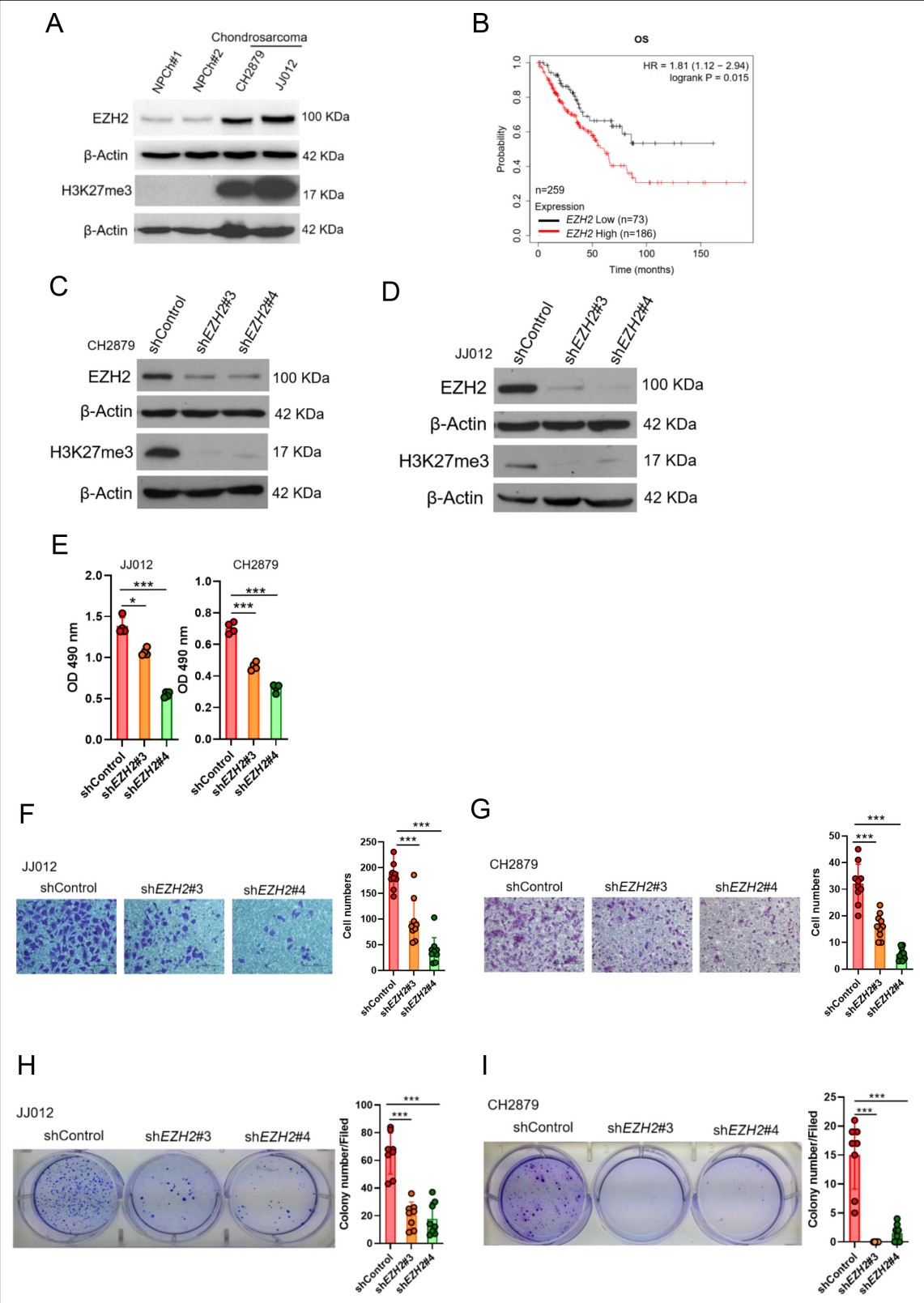

**Figure 1.** High *EZH2* expression predicts poor clinical outcome in chondrosarcoma patients. (**A**) Western blotting (WB) of EZH2 and H3K27me3 expression level in primary chondrocytes and chondrosarcoma cell lines. NPC, normal primary chondrocytes. (**B**) Prognostic correlation of survival analyses of sarcoma (including chondrosarcoma) patients with high and low *EZH2* level. OS, overall survival; HR, hazard ratio. WB of EZH2 and H3K27me3 expression level in CH2879 (**C**) and JJ012 (**D**) cell lines after knockdown (KD) of *EZH2* by different *EZH2* shRNA, respectively. (**E**) Proliferation

*Figure 1 continued on next page*

*Figure 1 continued*

of JJ012 and CH2879 cells was measured by MTS assay after depletion of EZH2. Migration assay of JJ012 (**F**) and CH2879 (**G**) cells was represented via using transwell while KD of *EZH2*. The quantification results were showed on the right panel. (**H and I**) Soft agar assay of JJ012 (**H**) and CH2879 (**I**) cells with control or various sh*EZH2* was conducted. The quantification results were represented on the right panel. Error bars represent mean ± SD (**E, F, G, H, and I**). $^*$p<0.05, $^{**}$p<0.01, $^{***}$p<0.001, and two-tailed unpaired *t* test (**E, F, G, H, and I**).

To further reduce the number of candidate regulators, we first focused on genes function as a tumor suppressor role among those downregulated by EZH2. This analysis resulted in seven known candidate genes (*Figure 2A*).

Of note, human sulftase1 (*SULF1*) was the top among those reduced expression in chondrosarcoma by EZH2 targeting. q-ChIP experiments further validated enrichment of EZH2 bound at Sulf1 promoter in various chondrosarcoma cell lines (*Figure 2B*). Consistently, qRT-PCR analysis showed that *SULF1* was downregulated in chondrosarcoma cell lines, JJ012 and CH2879 as compared with normal chondrocyte (*Figure 2C*). Immunoblotting also demonstrated that SULF1 expression was reduced in JJ012 and CH2879 cell lines (*Figure 2D*). Knockdown of *EZH2* induced SULF1 expression in both JJ012 and CH2879 cell lines (*Figure 2E and F*). The results indicated that *SULF1* is the direct targeted gene of EZH2 in chondrosarcoma cell lines. This regulation of EZH2/SULF1 axis was also observed in the other sarcoma cells such as osteosarcoma cell lines (*Figure 2—figure supplement 1A-G*). To further investigate the prognostic significance of EZH2/SULF1 axis in sarcoma, the Kaplan-Meier Plotter indicated that lower *SULF1* expression significantly correlated with poorer survival in sarcoma populations (*Figure 2G*).

## Ectopic expressed *SULF1* attenuates progression of chondrosarcoma

To further explore the potential effect of SULF1 on the chondrosarcoma progression, stably expressed *SULF1* of chondrosarcoma cell lines were established (*Figure 3A*), and assays for cell proliferation, migration, and colony-forming ability were conducted. *SULF1* ectopic-expression inhibited chondrosarcoma cell proliferation (*Figure 3B*), migration (*Figure 3C*), and colony-forming capacity (*Figure 3D*). Chondrosarcoma cells with ectopic-expressed *SULF1* were re-introduced control and specific SULF1 shRNA and followed by the test of colony-forming assay. The colony-forming capacity was restored in the presence of *SULF1* shRNA as compared with control (*Figure 3E*). We also examined the role of SULF1 in chondrosarcoma growth using an alternative tumor xenograft model. Subcutaneous injection of CH2879 cells resulted in reliable tumor growth in nude mice. The stable transfection of CH2879 cells with *SULF11* markedly repressed the growth of chondrosarcoma tumors with smaller size and weight compared with the control counterparts (*Figure 3F*). It is worth to mention that the *SULF1* level in the transfected chondrosarcoma cell lines are comparable to the expression level of *SULF1* in primary normal chondrocytes (n=107) by using qRT-PCR analysis (*Figure 3—figure supplement 1*). All these results revealed that SULF1 is a downstream target of EZH2 and serves as a tumor suppressor role in chondrosarcoma development.

## SULF1 repressed the signaling pathway of cMET via dissociation between of cMET and its co-receptor (HSPG) in chondrosarcoma

To understand the mechanism of SULF1-mediated inhibitory effects on the tumorigenesis of chondrosarcoma cell lines, the unbiased RTK antibody array was performed and revealed that several phosphorylation of RTKs were decreased in SULF1 overexpression cell lines, with the largest fold decreases in cMET, and followed by ALK and TIE2 (*Figure 4A and B*). To further confirm the phosphorylation of cMET in vector and SULF1 stably expressed chondrosarcoma cell line from antibody array, western blotting analysis was conducted, and all future analysis of site for phosphorylation of cMET refers to Y1234/Y1235, a catalytic activation site, unless otherwise specified. Furthermore, downstream signaling of cMET like AKT, ERK1/2, and p38 was also tested. Western blotting analysis indicated that in cells with ectopic expression of SULF1, phosphorylation of cMET, AKT, ERK1/2, and p38 was reduced (*Figure 4C*). Next, we assessed whether EZH2 enzymatic activity was required for SULF1 repression. The expression of SULF1 in chondrosarcoma cell lines treated with EPZ-6438 (tazemetostat), an EZH2 inhibitor which reduced the enzymatical activity of EZH2 without affecting its expression, was explored by western blotting. The expression of SULF1 was upregulated in a dose-dependent manner with EPZ-6438 (*Figure 4D*). Additionally, phosphorylation of cMET was reduced

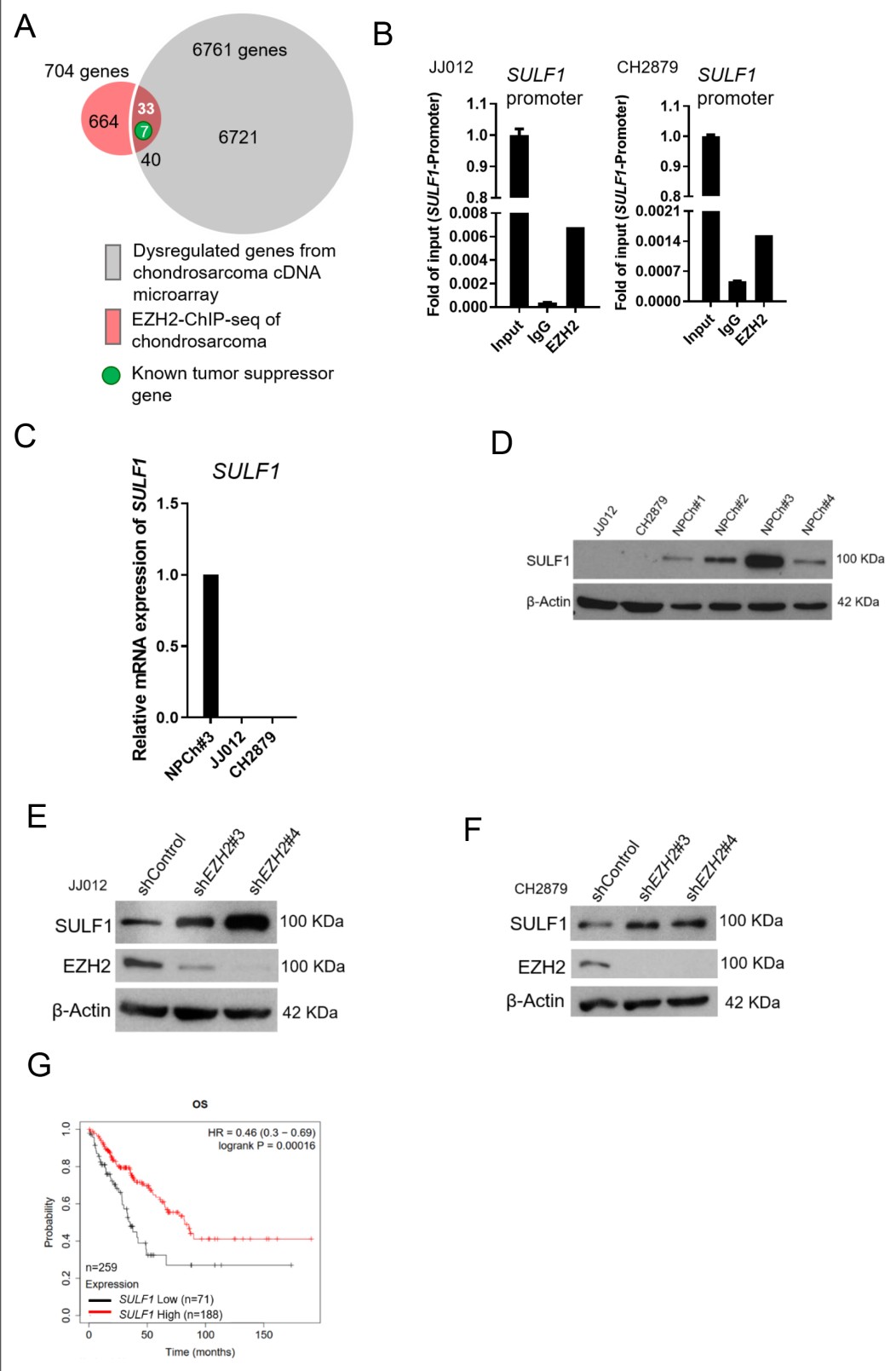

**Figure 2.** *SULF1* is the downstream targeted of EZH2 and repressed in chondrosarcoma cell lines. (**A**) cDNA microarray analysis was performed and compared between primary normal chondrocyte and JJ012 chondrosarcoma cell lines. Dysregulated genes were selected based on the criteria of 10-fold change. EZH2 chromatin immunoprecipitation sequencing (ChIP-seq) was conducted in JJ012 chondrosarcoma cell line.

*Figure 2 continued on next page*

*Figure 2 continued*

Venn diagram showing the overlap between genes (n=40) differentially dysregulated from cDNA microarray in chondrosarcoma cells compared to normal chondrocyte (n=6761) and genes targeted by EZH2 from EZH2 ChIP-seq in chondrosarcoma (n=704). (**B**) Quantitative chromatin immunoprecipitation (qChIP) assay of *SULF1* promoter was performed in JJ012 and CH2879 cells by using indicated antibodies. (**C**) Quantitative RT-PCR analysis of *SULF1* mRNA expression in normal chondrocyte, JJ012 and CH2879 cell lines. (**C**) EZH2 ChIP assay was performed in JJ012 and CH2879 cell lines using antibody against EZH2 or negative IgG control and analyzed by quantitative PCR. (**D**) Western blotting (WB) of SULF1 expression in normal chondrocytes and chondrosarcoma cells. WB of indicated protein in JJ012 (**E**) and CH2879 (**F**) cells harboring control or *EZH2* shRNA. (**G**) Prognostic correlation of survival analyses of sarcoma patients with high and low *SULF1* levels. OS, overall survival; HR, hazard ratio.

The online version of this article includes the following figure supplement(s) for figure 2:

**Figure supplement 1.** The regulation of EZH2/SULF1axis is also exhibited in osteosarcoma.

accompanied with the augmentation of SULF1 (*Figure 4D*). We also tested whether the loss of EZH2, which induces SULF1 expression, would impair cMET phosphorylation in chondrosarcoma cell lines. The phosphorylation of cMET was vigorously decreased while depletion of EZH2 in chondrosarcoma cell line (*Figure 4E*). These results suggest that EZH2 and its enzymatical activity is implicated in the regulation of cMET signaling pathway in chondrosarcoma.

Prior investigation reported that the SULF1 contain a large hydrophilic domain (HD), located between the N-terminal catalytic domain and C-terminal domain. The function of the HD of human SULF1 is governing enzyme activity, cell surface targeting, secretion, and substrate recognition. The double mutant SULF1 C87A/C88A was lacking catalytical activity (*Frese et al., 2009*) and serves as an enzymatically inactive SULF1 C87A/C88A (CA). To explore whether the phosphorylation of cMET is mediated by SULF1 enzyme activity, we generated stably expressed either wild type (WT) or CA mutant of *SULF1* in CH2879 cell line. The CA mutant of SULF1 lost enzymatic activity and was unable to repress the phosphorylation of cMET compared to the ectopic expressed WT SULF1 in CH2879 (*Figure 4F*), suggesting that sulfatase enzymatic activity of SULF1 was required for the dephosphorylation of cMET. Overexpression of WT SULF1 reduced the proliferation (*Figure 4G*) and colony forming ability in vitro (*Figure 4H*), as expected from the reduced phosphorylation of cMET in *Figure 4F*. In contrast, CA mutant SULF1 which lost enzymatic activity failed to do so in chondrosarcoma cell line (*Figure 4H*). To further evaluate the tumorigenicity of CA in vivo, chondrosarcoma stable cell lines with vector control, WT SULF1, and enzymatic inactive (CA) mutant SULF1 were subcutaneously injected into the nude mice. Tumor volume was measured twice per week. The results indicated that WT SULF1, associated with full sulfatase activity inhibited tumor development, consistent to what was observed in vitro culture. But the CA mutant, which lost sulfatase activity, thought to recover tumorigenicity, still associated with partial tumor suppression activity as it could not completely recover tumor volume similar to that of vector control (*Figure 4I*). To further elucidate the possible underlying mechanism for why the CA mutant which lost sulfatase activity still associated with partial tumor suppression activity. We examined the different phosphorylation site of cMET and found that CA mutant retained the similar phosphorylated profile of Y1234/1235 of cMET, a catalytic activation site, in cell lysates and tumors (*Figure 4—figure supplement 1A*). However, unexpectedly, the gain of phosphorylation of Y1003 of cMET was markedly detected in tumors, but not in the in vitro cell culture (*Figure 4—figure supplement 1B*). The phosphorylation of Y1003 of cMET has been implicated to associate tumor suppressive activity (*Organ and Tsao, 2011*). It is not yet clear how the tumor microenvironment might cause the unexpected results of CA mutant in the animal study (*Figure 4I*), and it is certainly interesting to pursue further how tumor microenvironment might affect p-Y1003 of cMET in the future. Additionally, higher cMET expression associated with poorer survival of patient with sarcoma was also observed (*Figure 4J*). To sum up, overexpression of WT SULF1 is implicated conferring decreased phosphorylation of cMET to attenuate cancer-related behavior in chondrosarcoma cell lines, and the enzyme activity of SULF1 is required for this action.

HSPGs act as co-receptors in cell signaling pathways and provide binding sites for growth factors and morphogens via specific sulfation patterns on their heparan sulfate (HS) chains. Through enzymatically removing 6-O-sulfate groups from HSPGs on the cell surface by SULF1, it can regulate the tricomplex formation among RTK, HSPGs and ligands, thereby modulating important processes such as development, cell growth, and differentiation (*Lai et al., 2008*). A high affinity ligand of cMET,

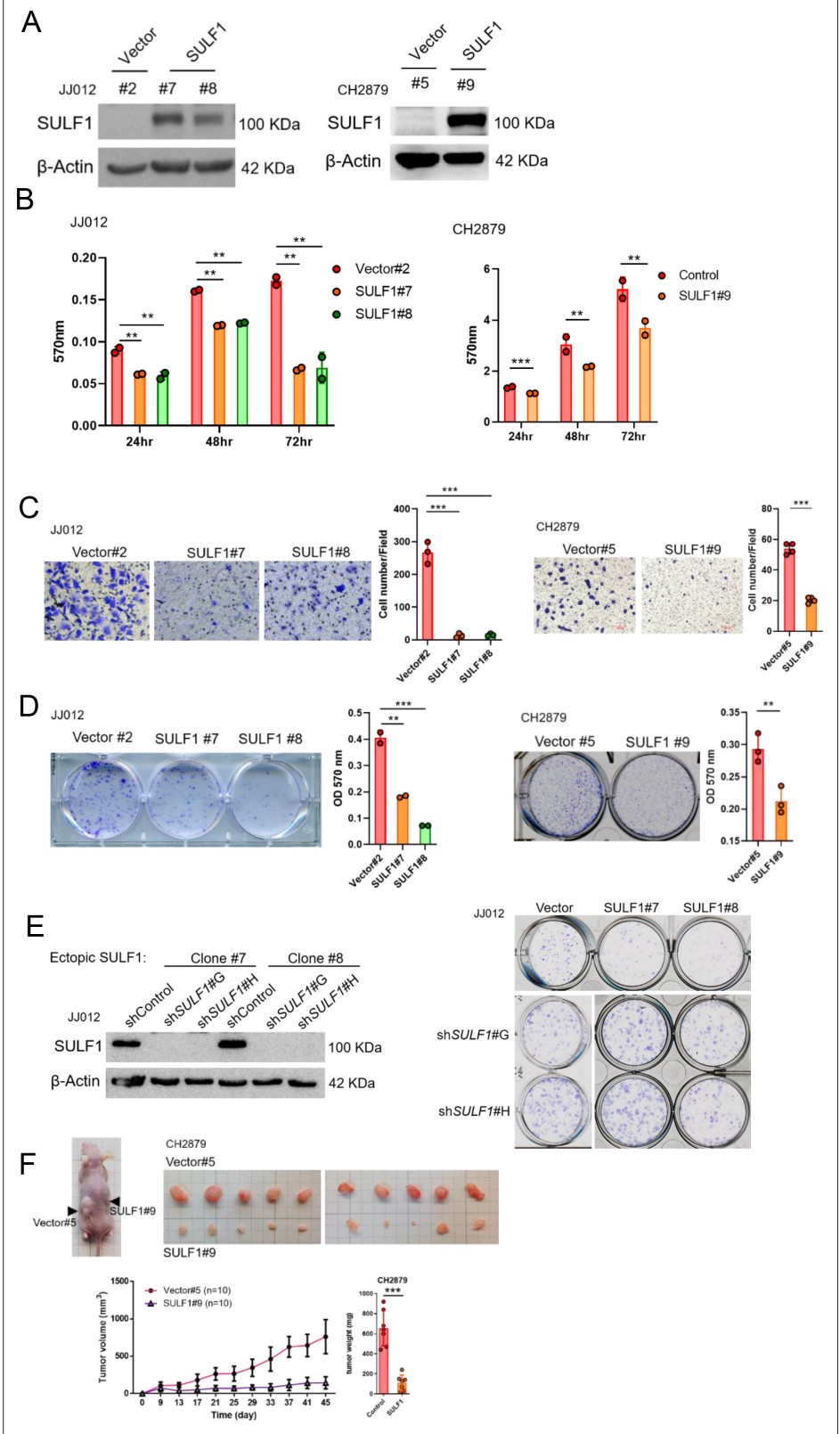

**Figure 3.** Ectopic expressed SULF1 attenuates tumorigenicity of chondrosarcoma. (**A**) Western blotting (WB) of SULF1 and vector control stable transfectants with SULF1 and β-actin antibodies. (**B**) MTT assay of JJ012 and CH2879 with SULF1 stable cell lines were performed in the indicated time point. (**C**) Quantification of migrated SULF1 stable cell lines of JJ012 (represented images, n=3) and CH2879 (represented images, n=4).

*Figure 3 continued on next page*

*Figure 3 continued*

(**D**) Quantification of colony forming assay of SULF1 stable cell lines including JJ012 (represented images, n=2) and CH2879 (represented images, n=3). (**E**) WB of ectopic expressed SULF1 stable cell line expressing indicated shRNA in left panel. Colony formation assay of SULF1 stable cell lines with indicated *SULF1* shRNA in the right panel. (**F**) BALB/c nude mice were subcutaneously injected vector (n=10) or SULF1 (n=10) stable cell lines, and tumor volume was showed at the indicated days after transplantation. Error bars represent mean ± SD (**B, C, D, and F**). *p<0.05, **p<0.01, ***p<0.001, and two-tailed unpaired *t* test (**B, C, D, and F**).

The online version of this article includes the following figure supplement(s) for figure 3:

**Figure supplement 1.** Quantitative RT-PCR analysis of *SULF1* transcript in primary chondrocytes (n=107) and SULF1 stable transfectants.

hepatocyte growth factor/scatter factor (HGF/SF) is well known to stimulate cMET activity and trigger important cellular processes including to enhance cell proliferation, invasion, survival, and angiogenesis. The HSPGs can bind to various growth factors and function as signal co-receptors. Actually, CD44 allows to increase the local concentration of glycosaminoglycan-associating proteins, such as osteopontin (OPN), FGF2, VEGF, and HGF, thereby inducing the capacity of these ligands to interact with their receptors and so reducing the threshold for signal transduction (*Lai et al., 2009*; *Thayaparan et al., 2016*). Interaction of CD44 and cMET facilitated tumor cell migration, invasion, and metastasis (*Jeon and Lee, 2017*; *Zoller, 2011*). Most of these interactions induce synergistic effects on cancer progression and the generation of resistance to therapy (*Engelman et al., 2007*; *Viticchiè and Muller, 2015*).

Thus, we hypothesized that SULF1 may modulate cMET signaling by desulfating cell surface HS glycosaminoglycans (HSGAGs) and thus abrogating HSGAG-dependent growth signaling. A model was shown in Figure 8D.

To further test this hypothesis, the phosphorylation of cMET was examined by western blotting in vector and ectopic SULF1 expressed chondrosarcoma cell lines treated with or without HGF ligand. The phosphorylation of cMET was dampened in SULF1 overexpressed chondrosarcoma cell lines (*Figure 5A and B*). The interaction between cMET and CD44 was clearly detected by immunoprecipitation analysis in vector control chondrosarcoma cell lines as reported, yet such interaction between cMET and CD44 was abolished in SULF1 overexpressed chondrosarcoma cell lines (*Figure 5C*). To clarify whether SULF1 expression leads to desulfation of cell surface HSGAG, we performed flow cytometry and western blot utilizing the 10E4 anti-HSPG antibody to recognizes N-sulfated glucosamine residues. The ectopic expressed SULF1 of chondrosarcoma cell lines showed the cell surface staining for N-sulfated glucosamine containing HSGAGs was diminished compared with vector control chondrosarcoma cell lines via using flow cytometry analysis (*Figure 5D and E*). Furthermore, western blot of sulfated HSGAGs in SULF1 ectopic expressed cell line was reduced in contrast to the counterpart (*Figure 5F*). Together, these results indicated that downregulation of SULF1 enhances the complex formation of cMET, HGF, and CD44, consequently to increase the downstream survival signaling to promote tumorigenesis of chondrosarcoma.

## Treatment with cMET pharmaceutical inhibitors effectively blocks chondrosarcoma growth and prolongs mice survival

We, therefore, determined whether pharmacological inhibition of EZH2 or cMET activity could be an option for target therapy for chondrosarcoma. EZH2 inhibitor such as GSK343 and EPZ-6438 (tazemetostat), which compete with sadenosyl-methionine for binding to EZH2, thereby inhibiting histone methyltransferase activity without affecting EZH2 protein expression was examined. The cMET small molecule inhibitors, tivantinib, capmatinib, and crizotinib, were also used to test the treatment efficacy. In contrast to cMET inhibitors, EZH2 inhibitor required higher IC50 concentrations (*Figure 5—figure supplement 1*) to diminish trimethylation of histone lysine 27 (H3K27me3; *Figure 5—figure supplement 1*) and colony-forming ability (*Figure 5—figure supplement 1D, E*) in chondrosarcoma cell lines. For cMET inhibitors, the IC50 of campatinib (12 µM) was much higher than tivantinib (1 or 1.2 µM) and crizotinib (0.5 µM) in chondrosarcoma cell lines (*Figure 6—figure supplement 1*). Tivantinib, crizotinib, and capmatinib effectively attenuated the phosphorylation of cMET in both CH2879 and JJ012 chondrosarcoma cell lines (*Figure 6A*). Particularly, tivantinib and crizotinib decreased the phosphorylation of cMET with low dosage at 0.4 and 0.3 µM, respectively (*Figure 6B*). Thus, we

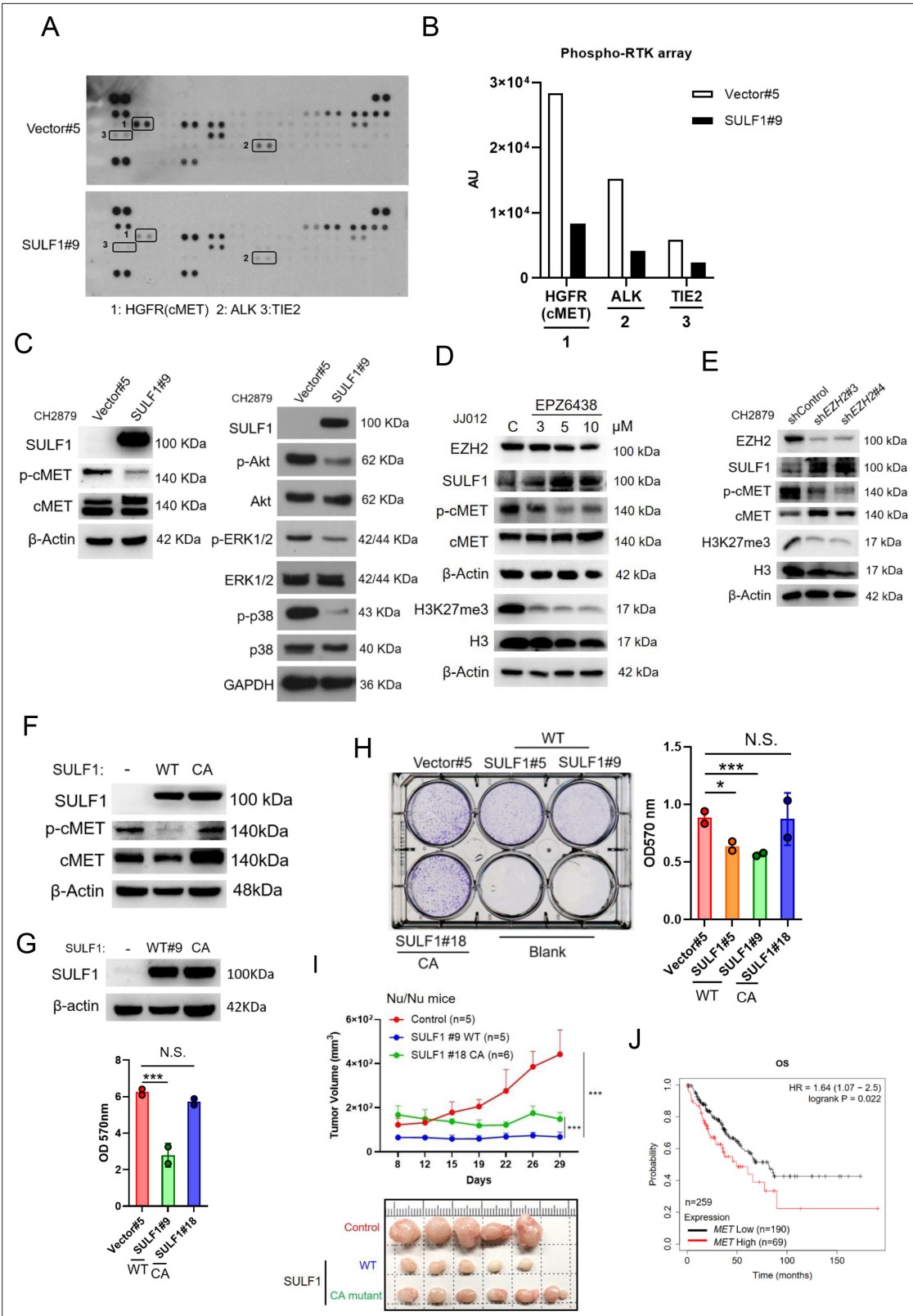

**Figure 4.** SULF1 mediates cMET signaling and is required its enzymatic activity. (**A**) Human phosphor-RTK array analysis of CH2879 vector control and SULF1 stable cell line. Three pairs of positive signals in duplicate coordinates (vector compare to stable cell line) are shown in HGFR (C3/C4), ALK (E13/E14) and Tie-2 (D1/D2). (**B**) The quantification was represented, and the signals were detected by ImageJ. AU: arbitrary unit. (**C**) Western blotting (WB) of SULF1, phosphorylation of cMET and cMET expression in CH2879 vector and SULF1 stable cell line (left). The RTK downstream protein expression

*Figure 4 continued on next page*

*Figure 4 continued*

by western blot analysis (right). (**D**) WB of EZH2, H3K27me3, histone3, phosphorylation of cMET and MET expression in JJ012 treated with EPZ-6438. (**E**) WB of EZH2, phosphorylation of cMET and MET expression in CH2879 in depletion of EZH2. (**F**) WB of phosphorylation level of cMET in SULF1 wild type (WT) and enzymatic inactive mutant (CA) stable cell lines. (**G**) Top, WB of SULF1 expression level in WT SULF1 and mutant (CA) stable cell lines. Lower, MTT assay of WT and CA SULF1 stable transfectants. (**H**) Colony-forming assay of vector control, WT and CA SULF1 stable cell lines. Quantification data was showed in right panel. (**I**) BALB/c nude mice were subcutaneously inoculated stable cell lines with vector (n=5), WT SULF1(WT; n=5), and enzymatic inactive SULF1 (CA mutant; n=6), tumor growth was plotted at the indicated days after transplantation. Error bars represent mean ± SD. (**J**) Kaplan-Meier plot of overall survival of the patients stratified by high and low *cMET* level. Error bars represent mean ± SD (**G, H, and I**). $^*p<0.05$, $^{**}p<0.01$, $^{***}p<0.001$, and Two-tailed unpaired $t$ test (**E and F**).

The online version of this article includes the following figure supplement(s) for figure 4:

**Figure supplement 1.** Phosphorylation of cMET at the indicated site in chondrosarcoma cell lines and tumors with stably expressed vector control, wild type (WT) SULF1, and CA mutant SULF1.

focused on tivantinib and crizotinib inhibitors for the further assays. Reduction of colony forming capacity was also observed in both CH2879 and JJ012 cell lines by treating with tivantinib and crizotinib (*Figure 6C and D*). Collectively, the treatment efficacy of cMET inhibitors is effectively to reduce colony formation in chondrosarcoma cell lines.

Next, we explored the effects of pharmacologically targeting cMET on tumor growth in mice. Mice with established luciferase-CH2879 tumors were treated with vehicle or crizotinib (*Figure 7A*). Crizotinib treatment inhibited cartilage tumor growth in orthotopic immunodeficiency mice in a dose-dependent manner (*Figure 7B and C*) without altering the mice body weight during administrated vehicle or crizotinib (*Figure 7D*). Moreover, western blotting was shown that crizotinib therapy could decrease the phosphorylation of cMET in mice tumors (*Figure 7F*), which results in attenuating chondrosarcoma progression and dramatically ameliorate survival (*Figure 7G*). Overall, these results demonstrated that crizotinib had therapeutic implications to inhibit the growth of chondrosarcoma in mice.

## Clinical significance of the SULF1/cMET pathway in chondrosarcoma

To further validate our observation in clinical setting, we examined the clinical significance of EZH2, SULF1, and phospho-cMET by analyzing H-score of immunohistochemistry (IHC) on commercially available human bone cancer tissue array. As a result, the patient samples exhibited a low-level expression of SULF1 and high-level expression of EZH2, and phospho-cMET compared to their counterpart (*Figure 8A-C*). These similar results were also observed in osteosarcoma tissue samples (n=31; *Figure 8—figure supplement 1*). Together, our data demonstrate that EZH2-mediated downregulation of SULF1 is critical for de-suppressed cMET signaling pathway in bone cancer. In chondrosarcoma, the expression of EZH2 was upregulated and thereby inhibited the SULF1 expression. Reduction of SULF1 cannot remove the 6-O sulfation in HSPG resulting in the stabilization of HGF, cMET, and CD44 tricomplex to activate the downstream signaling such as phosphorylation of AKT, p38, and ERK (*Figure 8D*).

## Discussion

Chondrosarcomas are heterogeneous group of malignant cartilaginous neoplasms with various morphological features, represented by resistance to chemo and radiation therapies. Accumulating molecular alteration involved in chondrosarcoma pathogenesis has been characterized in the last several decades (*Amary et al., 2011*; *Tarpey et al., 2013*; *Totoki et al., 2014*; *Zhang et al., 2013*); however, no FDA-approved targeted therapies are currently available for chondrosarcoma (*Monga et al., 2020*). Hence, the identification of novel targets for new treatment options remains urgent. Here, we reported that expression of EZH2 was augmented in chondrosarcoma cell lines, which exerted a critical role in tumor progression. Over the past years, EZH2 inhibitors have been generated and tested. The compound, DZNep, was shown to attenuate EZH2 protein expression and subsequently decreased H3K27me3, resulted in promoting cell death of chondrosarcoma cell line by apoptosis (*Girard et al., 2014*). Compared to our results, selective EZH2 inhibitor, GSK343, and the FDA-approved EPZ-6438 (tazemetostat) were utilized to treat chondrosarcoma cell lines on colony-forming ability assay. The inhibitory effects of colony formation of chondrosarcoma cell lines

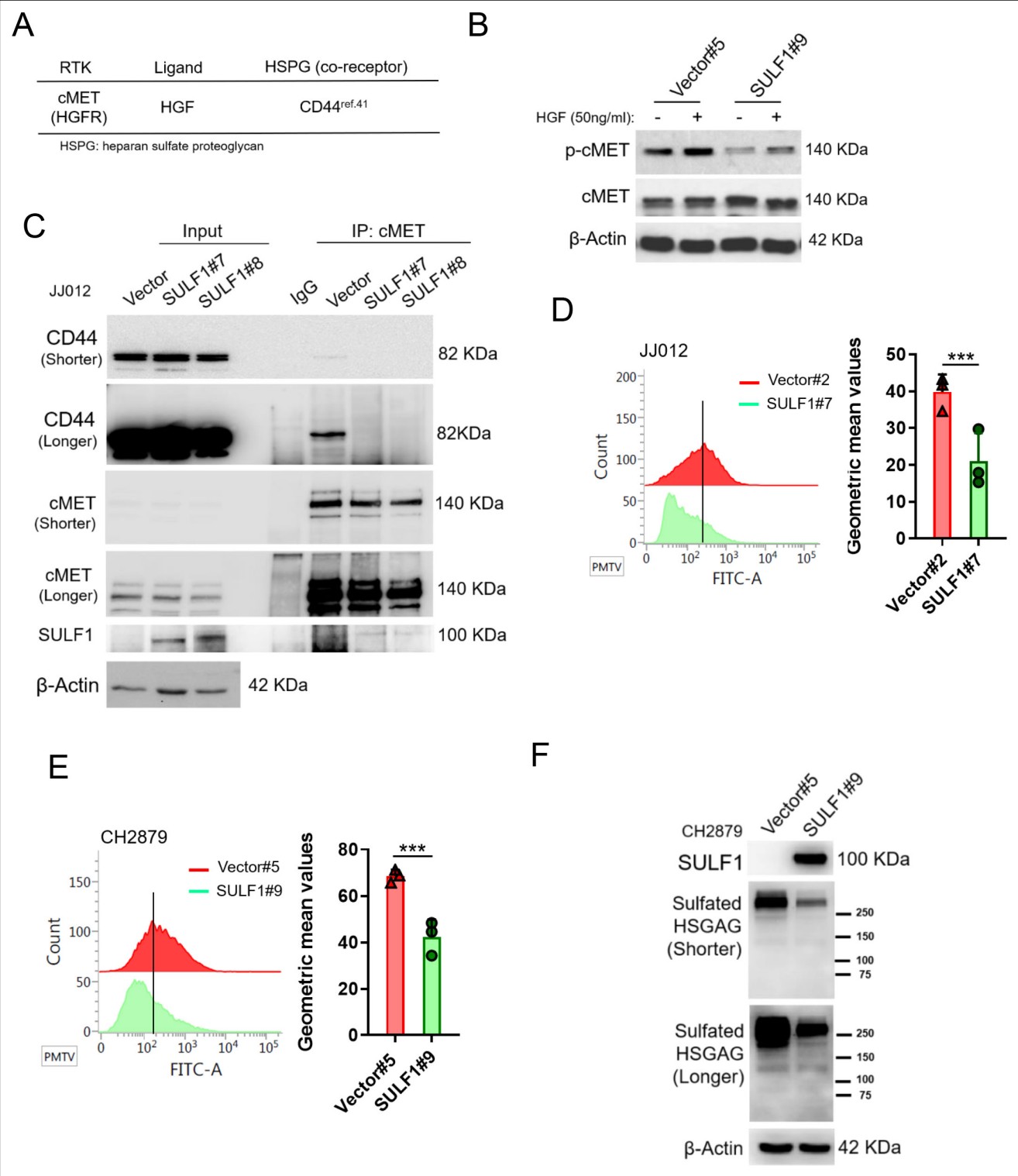

**Figure 5.** SULF1 reduces the interaction between cMET and its co-receptor, CD44, by removing the sulfate group of CD44. (**A**) cMET and its putative co-receptor was showed in table. (**B**) Western blotting (WB) of phospho-cMET and cMET expression in generated vector and SULF1 stable transfectants treated with or without hepatocyte growth factor (HGF). (**C**) Immunoprecipitation (IP) assay with IgG or cMET antibodies of JJ012 cells ectopically expression SULF1 or vector control, followed by WB of indicated antibodies. Flow cytometry analysis of sulfated heparan sulfate glycosaminoglycan (HSGAG) by anti-HSGAG monoclonal antibody 10E4 on cell surface of JJ012 (**D**), and CH2879 (**E**) of the SULF1 stable clones and the counterparts. (**F**) WB of sulfated HSGAG in vector and SULF1 stable transfectants with 10E4 antibody. Error bars represent mean ± SD (**D and E**). *p<0.05, **p<0.01, ***p<0.001, and two-tailed unpaired *t* test (**D and E**).

*Figure 5 continued on next page*

*Figure 5 continued*

The online version of this article includes the following figure supplement(s) for figure 5:

**Figure supplement 1.** IC50 value of EZH2 inhibitors were examined by MTT assay in CH2879 cells (**A**), JJ012 cells (**B**).

by GSK343 (5–10 µM; Supplementary Fig. 4) exhibited lower efficiency in contrast to the crizotinib (0.2–1 µM) treatment in vitro (*Figure 6*). It seems that the FDA-approved cMET inhibitors showed more potent retardation on chondrosarcoma tumorigenesis than inhibition of currently available EZH2 inhibitors.

Sulfatase can hydrolyze sulfate ester bonds of a broad range of substrates to remove 6-O-sulfate groups including HSPGs, which acts as a co-receptor for various heparin-binding growth factors and cytokines subsequent altered signaling pathways. Delineation of SULF1 expression in distinct epithelial cancer types has revealed the expression was completely lacked or markedly decreased in various cancer cell lines (*Lai et al., 2004b*; *Lai et al., 2008*), suggesting that reduction of SULF1 is a general feature among epithelial cancers. Nevertheless, the role of SULF1 in chondrosarcoma is unclear. Mounting references showed the heparin-degrading endosulfatase SULF1 in cancers may contribute to affect cellular growth and survival signaling, tumor proliferation, migration, invasion, and angiogenesis in vitro and in vivo. Thus, to desulfate HSPGs by SULF1 might have a tumor suppressor effect via abolishment of ligand-receptor binding and downstream signaling. Prior studies revealed that the augmentation of 6-O-sulfation via three HS 6-O-sulfotransferases (HS6ST1-3) may promote the cartilage tumor progression. Simultaneously, they also showed that expression of SULF1 did not exhibit significant difference between benign tumor and distinct histological grade of chondrosarcoma (*Waaijer et al., 2012*). In contrast, we observed the inhibition of SULF1 expression through histone 3 K27 trimethylation by EZH2 to attenuate the elimination of 6-O-sulfation in chondrosarcoma, resulting in the high level of 6-O-sulfation of HSPGs to enhance RTK signaling like cMET. Evidence has been demonstrated that downregulation of SULF1 in multiple cancers including ovarian, breast, gastric, and hepatoma is orchestrated by DNA hypermethylation of CpG islands which located within promoter or exon 1 A to prevent the transcription factors from binding to their DNA binding sites (*Lai et al., 2004b*; *Lai et al., 2004a*; *Staub et al., 2007*), concluding that epigenetic modulation was crucial for SULF1 silencing in cancer cells.

Here, we showed that cMET phosphorylation was decreased in SULF1 overexpressed cells by using phospho-proteomic analysis of RTKs in vector and SULF1 ectopic expressed cell lines, suggesting that SULF1 mediated the activity of RTKs. cMET has been reported to facilitate migration of chondrosarcoma cell lines upon HGF interaction (*Tsou et al., 2013*). Also, cMET signaling contributes to tumor survival, growth, angiogenesis, and metastasis, and several cMET inhibitors were approved by FDA for cancer treatment. Prior investigation demonstrated that CD44 was highly expressed in surface of chondrocytes and was a receptor for hyaluronan. Moreover, it binds to HGF and cMET to form complex to induce oncogenic signaling. Numerous post-translational modifications including N- and O-linked glycosylation, phosphorylation, sulfation, and domain cleavage, which shapes the complication and function of CD44 protein family. For instance, the proinflammatory cytokine like tumor necrosis factor-α (TNF-α) was illustrated to change the inactive form of CD44 into active form via induction of the sulfation of CD44 (*Maiti et al., 1998*). Consistently, we showed that the reduction of the sulfation of CD44 by SULF1 is leading to inactivate the HGF, cMET, and CD44 tricomplex and repressed its downstream signaling.

Although there is no FDA approval targeted drug for chondrosarcoma, plenty of drugs are tested in vivo studies and clinical trials. Previously, various groups attempted to identify appropriate targets for prediction and therapies of chondrosarcoma by using diverse approaches. For instance, *Nicolle et al., 2019* indicated that integrated multi-omics classification to reveal the significance of the loss of expression of the 14q32 locus in defined malignancy on cartilage tumors. The IDH activating mutations driving a genome-wide hypermethylation in chondrosarcoma contribute to the activation of proliferative and glycolytic state, which was potentially driven by the hypoxia inducible factor (*Nicolle et al., 2019*). The other group uncovered that HIF-2α as a critical regulator for tumor growth, local invasion, and metastasis in chondrosarcoma by using weighted gene co-expression network analysis (*Kim et al., 2020*). They also demonstrated that combination with small molecule HIF-2α inhibitor and chemotherapy agents synergistically promoted chondrosarcoma cell apoptosis and abrogates the malignant features of chondrosarcoma in mice.

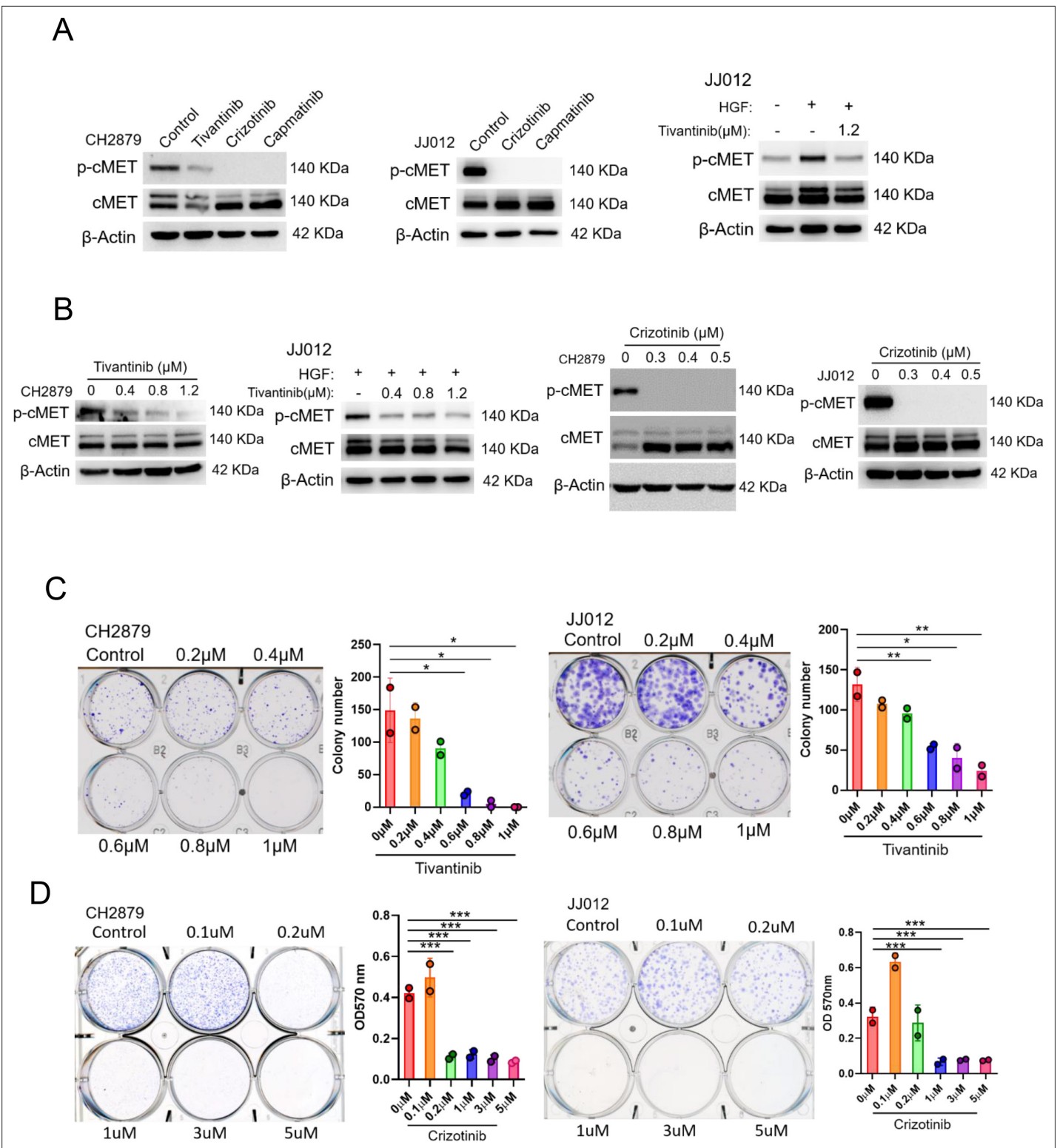

**Figure 6.** cMET inhibitors attenuate phosphorylation and colony-forming ability of chondrosarcoma cell lines. (**A** and **B**) Western blotting (WB) of phospho-cMET and cMET in CH2879 and JJ012 cell lines treated with tivatinib, crizotinib, and capmatinib at indicated dosage. JJ012 cells were pre-treated with 50 ng/ml human recombinant hepatocyte growth factor (HGF) for 30 min before treated with tivantinib. (**C** and **D**) Colony formation assay of CH2879 and JJ012 cell lines with or without indicated inhibitors. Quantitation of colony formation assay by CH2879 (represented images, n=2), JJ012 (n=2) cells. Error bars represent mean ± SD (**C** and **D**). *p<0.05, **p<0.01, ***p<0.001, and two-tailed unpaired *t* test (**C** and **D**).

The online version of this article includes the following figure supplement(s) for figure 6:

*Figure 6 continued on next page*

*Figure 6 continued*

**Figure supplement 1.** IC50 value of cMET inhibitors were examined by MTT assay in CH2879 cells (**A**), JJ012 cells (**B**).

Recently, Palubeckaite and colleagues have generated an alginate-based 3D cell culture model of chondrosarcoma to evaluate the treatment of chemotherapeutic drugs and target therapeutic agents including Sapanisertib (mTOR inhibitor) and AGI-519 (inhibitor of mutant IDH1). They reported that more resistant results of those treatment were observed in 3D compared to 2D cell culture system (*Palubeckaitė et al., 2020*). Inhibition of mTOR alone in this 3D system was showed lower effectiveness and which might be improved by combination treatment to eliminate the remining cell population (*Boehme et al., 2018*). High expectations for effective treatment was occurred while the characterization of specific driver mutations in the IDH genes IDH1 and IDH2, and a specific inhibitor of mutant IDH1 was developed such as AGI-5198. Previous investigations reported that using 3D culture model was reduction of clonogenic capacity only after long-term treatment under high concentrations, yet no benefit of treatment was detected at lower concentrations or shorter treatment times (*Li et al., 2015*; *Suijker et al., 2015*). Improved version of IDH mutation inhibitor, ivosidenib (AGI-120), increased the metabolical stability and was approved for chondrosarcoma clinical trial (*Popovici-Muller et al., 2018*). However, it was demonstrated poor progression-free survival with only one agent therapy in the retrospective investigation (*van Maldegem et al., 2019*). Suggesting that the limiting effects on the target therapies in chondrosarcoma thus far, tivantinib (ARQ-197) is a potent non-ATP competitive selective and oral c-Met small molecule inhibitor currently under evaluation of clinical trial for liver and lung cancers (*Michieli and Di Nicolantonio, 2013*). Here, we also showed its potent inhibitory effect on phosphorylation of cMET and colony-forming ability in chondrosarcoma cell lines. Compared to cMET inhibitors including crizotinib, capmatinib, and tivantinib, tivantinib was more specific for inhibition of MET (*Vansteenkiste et al., 2019*), yet chondrosarcoma seems to require higher dose of IC50. Crizotinib is a type Ia tyrosine kinase inhibitor that is approved for the treatment of ALK or ROS1-rearranged advanced non-small cell lung cancers (*Goździk-Spychalska et al., 2014*). In addition to its activity against ALK and ROS1, it has potent activity against cMET and low nanomolar potency in cell lines including our studies. It is not yet clear why crizotinib showed a better IC50 than capmatinib, one possibility might be due to the ability to target multiple RTKs. It is certainly worthy of pursuing further the underlying regulatory mechanisms.

We demonstrated that EZH2/SULF1 axis mediates cMET in chondrosarcoma, and inhibition of cMET at low dose for long-term administration effectively alleviates tumor growth of chondrosarcoma and prolongs survival in mice. A effective treatment by crizotinib in chondrosarcoma. Targeting EZH2/SULF1/cMET shown in this report provides proof of concept evidence to use cMET as biomarker to select chondrosarcoma patients for treatment by the FDA-approved cMET inhibitors such as crizotinib.

## Materials and methods
### Cell culture

Human chondrosarcoma cell line, JJ012, was provided from Dr. Joel Block and maintained in Dulbecco's Modified Eagle's Medium - high glucose (DMEM, Gibco), F12, minimum essential medium (MEM, Gibco), 10% fetal bovine serum (FBS) and 1% P/S in a ratio of 40: 10: 40: 10: 1%, supplemented with insulin, ascorbic acid, and hydrocortisone. CH2879 was kindly provided from Dr. Antonio Llombart-Bosch and cultured in RPMI1640 (Gibco) contained 10% FBS and 1% P/S. Patients-derived tissues were provided by Dr. Teng-Le Huang (CMUH IRB No.102-REC1-047-AR) and isolated (*Goldring, 1996*). Based on the agreement of informed consent form, the cartilage tissues were obtained from the patients with surgery of total knee replacement. The results of studies would be allowed to publish with patient information confidentiality, which was also included in the consent form. Normal primary chondrocytes were culture with DMEM/F12 contain 1% P/S/F and 10% FBS. Human osteosarcoma cell line, MG63, was provided form Dr. Shian-Ying Sung and cultured in Dulbecco's Modified Eagle's Medium-high glucose (DMEM, Gibco) contain 10% FBS and 1% P/S. U2OS was provided form Dr. Jer-Yuh Liu, and cultured in McCoy's 5 A medium contain 10% FBS and 1% P/S. All cell lines were negative for *mycoplasma* contamination tests and validated by the short tandem repeats (STRs) sequencing (Mission Biotech; Taipei, TAIWAN).

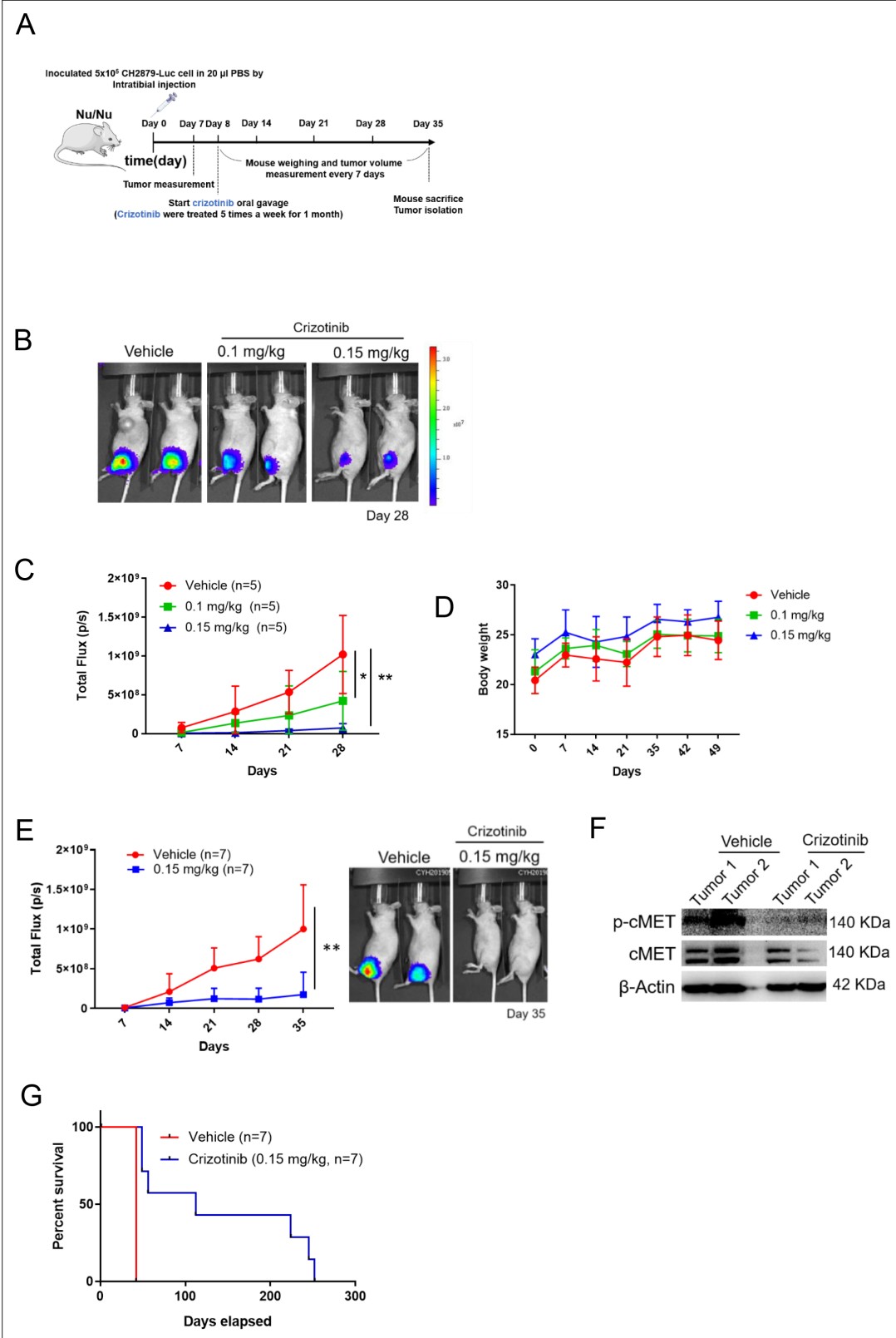

**Figure 7.** cMET inhibitor decreases tumor growth and prolongs mice survival. (**A**) Schematic illustration of treatment with crizotinib in orthotopic xenograft of CH2879 cells. (**B**) Mice were oral gavage with vehicle or crizotinib (n=5). Representative bioluminescent images of mice in following treatment at day 28. (**C**) Growth curves of tumors in immunodeficient mice (n=5) for days 7, 14, 21, and 28. (**D**) Curves of body weight of mice treat with or without crizotinib at indicated days. (**E**) Representative quantification and images of bioluminescent of mice in following treated with vehicle (n=7) or

*Figure 7 continued on next page*

*Figure 7 continued*

0.15 mg/kg (n=7) for mice survival. (**F**) Western blot analysis for phosphor-cMET of tumors isolated from mice. (**G**) Survival curves for mice from E. Error bars represent mean ± SD (**C and E**). *p<0.05, **p<0.01, ***p<0.001, and two-tailed unpaired *t* test (**C and E**).

## Real-time RT-PCR

Total RNA will be isolated using Trizol (Invitrogen) according to the manufacturer's instructions. After total RNA isolation, cDNA was generated by SuperScript TM III reverse transcriptase (Invitrogen) and analyzed by real-time PCR (Roche, LightCycler 480) with SYBR Green. All reaction products were normalized to the mRNA expression level of *ACTB.*

## Quantitative ChIP (qChIP) assay and ChIP-seq

ChIP assays were performed by EZ-ChIP kit (#17–371, Upstate) according to the manufacturer's instructions. EZH2 antibodies (#5246 Cell Signaling) were used for immunoprecipitation in the ChIP assays. The immunoprecipitated DNA was subjected to real time PCR by Roche Cybr Green system according to the manufacturer's instructions. For ChIP-seq analysis, the immunoprecipitated DNA was sequenced by Center for Genomic Medicine in NCKU (National Cheng Kung University), and sequenced data was further analyzed through NCKU bioinformatics center.

## Lentiviral infection

The lentiviral shRNA (target sequence: sh*EZH2*-3: CGGAAATCTTAAACCAAGAAT; sh*EZH2*-4: GAAA CAGCTGCCTTAGCTTCA; sh*SULF1*-#7: GCGAGAATGGCTTGGATTAAT; sh*SULF1*-#8: GCCGACCA TGGTTACCATATT) clones were purchased from National RNAi Core Facility of Academic Sinica, Taiwan. Cells were infected with vector alone (pLK0.1) lentivirus or indicated shRNA lentivirus. Media containing lentivirus was collected and used to infect target cells supplemented with Polybrene (8 µg/ml).

## Immunoblotting and immunoprecipitation

Cells were harvested on ice by using NETN (150 mM NaCl, 1 mM EDTA, 20 mM Tris-HCl, and 0.5% NP40) buffer supplemented with protease inhibitors, and lysates were incubated with antibodies for overnight at 4°C. Immunocomplexes were precipitated with protein A/ G (Roche Applied Science) for 3 hr at 4°C. The beads were washed four times with NETN buffer. Proteins were then eluted from the beads by boiling in sample buffer for 5 min and then analyzed by electrophoresis on SDS-polyacrylamide gels and transferred to polyvinylidene difluoride membranes. Whole-cell lysates were analyzed with the following antibodies: pcMET (#3077 Cell Signaling), cMET (#8198 Cell Signaling), CD44v6 (BBA13 R&D), pERK 1/2 (GTX59568 #9101 Cell Signaling), ERK 1/2 #4695 Cell Signaling, H3K27me3 ab6002 Abcam, Histone 3 (GTX122148, #9715 Cell Signaling), ß-Actin (sc-47778 Santa Cruz Biotechnology), EZH2 (#3147, #5246 Cell Signaling), pP38 (#9211 Cell Signaling), P38 (sc-7149 Santa Cruz Biotechnology), SULF1 (ab32763 Abcam), and 10E4 (#370255 S, amsbio). The image detection was analyzed by ChemiDoc Touch Imaging System (Bio-Rad) and Image Lab software (Bio-Rad).

## Proliferation assay

The proliferative ability of JJ012 or CH2879 cells with or without *EZH2* shRNA were seeded at a density of $3 \times 10^3$ cells or $5 \times 10^3$ cells/well in a 96-well tissue culture plate, respectively. Cells density were measured via Cell titer 96 one aqueous cell proliferation kits (MTS assay) after seeding for 48 hr by ELISA reader at OD 490 nm. Each treatment was tested in quadruplicate. The same experiment was independently repeated three times. The data were evaluated against shRNA control using Student's *t* test. The proliferation of vector and SULF1 ectopic expressed JJ012 and CH2879 stable clones were seeded $2 \times 10^4$ cells and $1.1 \times 10^5$ cells per well in 6-well plate, respectively. The cells were incubated for 24, 48, and 72 hr. After emptied culture medium, 1 mg/ml MTT (3-[4, 5-ciemthylthiazol-2-yl]–2, 5-diphenyltetrazolium bromide, Cat. No. M6494, Invitrogen) was added to each well for 1 ml and followed by incubated for 4 hr in the dark under humidified conditions at 37°C and 5% $CO_2$. Excess MTT was removed and then added 1 ml DMSO to each well. The plate was shacked for 5 min, and they were detected with OD 570 nm by ELISA reader. For $IC_{50}$, JJ012 or CH2879 cells were seeded at $7 \times 10^4$ or $9 \times 10^4$ cell density into 24 well, respectively. Cells were treated with different

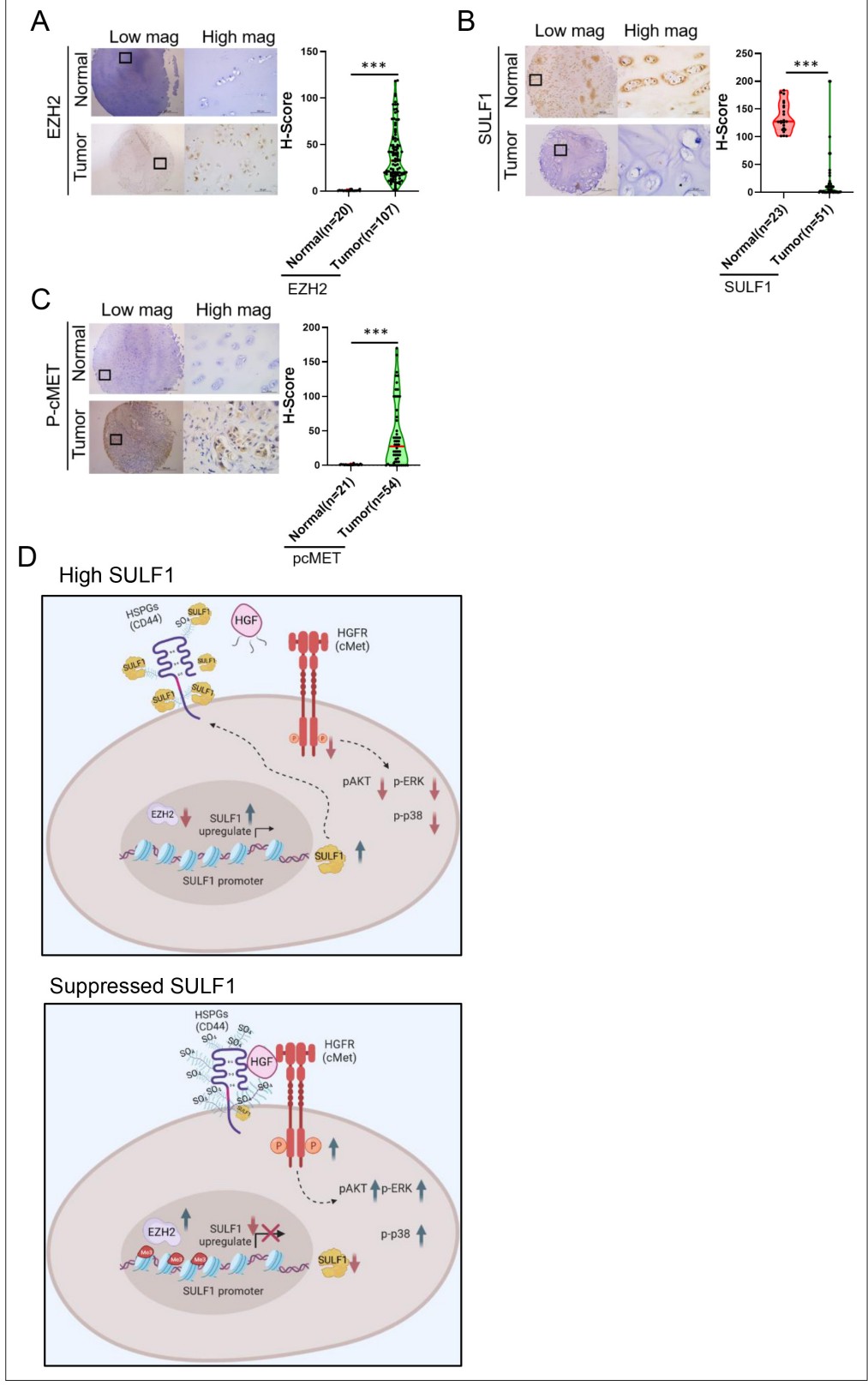

**Figure 8.** Pathological relevance of EZH2/SULF1/cMET axis in chondrosarcoma. (**A–C**) The representative cases of immunohistochemistry (IHC) staining for EZH2, SULF, and phospho-cMET expression in human paraffin embedded chondrosarcoma tissue array. Quantification of IHC staining by H-score via Image Scope software. (**D**) A propose model of the regulation of EZH2/SULF1/cMET axis in chondrosarcoma. Briefly, SULF1 expression was suppressed

*Figure 8 continued on next page*

*Figure 8 continued*

by EZH2, consequently de-reduced the sulfation of CD44 through the downregulate of SULF1. This de-reduction of sulfate group increases the complex formation of HGF, cMET, and CD44, thereby enhancing and trigging the downstream signaling of cMET. Artwork by Y-C. L., Z-S.L., and Y-H. (**C**) was created with BioRender.com.

The online version of this article includes the following figure supplement(s) for figure 8:

**Figure supplement 1.** The representative cases of immunohistochemistry (IHC) staining for EZH2, SULF1, and phospho-cMET expression in human paraffin embedded osteosarcoma tissue array (n=31).

concentrations of cMET (Capmatinib, Selleckchem- INCB28060, Crizotinib, Sigma-PZ0191; Tivantinib, Selleckchem-S2753) and EZH2 (EPZ-6438, MCE-HY-13803; GSK343, Sigma-SML0766) small molecular inhibitors for 48 hr. The cell viability was determined by MTT assay. IC50 values were calculated by concentration-response curve fitting using a four-parameter analytical method via PRISM software.

## Migration assay

The migration assay was performed using the transwells (CORstar COR-3422 pore size, 8 μm) in a 24-well plate. The bottom chambers were filled with cultured media containing final concentrations of 10% heat-inactivated FBS. JJ012 and CH2879 cells at a density of $1 \times 10^4$ cells in serum free cultured media were added to upper chambers. After incubation at 37°C for 16 hr, the membrane was fixed with methanol and stained with crystal violet then analyzed for the number of stained cells that had migrated to the opposite side of the membrane.

## Soft agar assay

The 6-well plate was added with 1 ml 0.5% agarose. The $5 \times 10^3$ cells of JJ012 or CH2879 were mixed with a 1 ml culture medium containing 0.3% agarose after the agar solidified. Each well was supplemented with culture medium and maintained until colony formation. For colony quantification, the colonies were stained by 1% crystal violet and analyzed.

## Clonogenic assay

For knockdown of SULF1 in JJ012 ectopic expressed SULF1 stable cell line, cells were seeded 750 cells per well into a 12-well plate; Tivantinib treatment, JJ012 and CH2879 cells were seeded 500 cells into a 12-well plate; Crizotinib treatment, JJ012 and CH2879 cells were seeded $1 \times 10^3$ cells per well; CH2879 vector, WT SULF1, and SULF1 enzymatical inactivation (CA mutant) stable cell lines, $3 \times 10^3$ cells per well were seeded into a 6-well plate. All cells were incubated for 9–11 days. Once colony formed, the plates were washed with PBS once and fixed with 75% methanol containing 25% acetic acid solution at room temperature for 5 min. After fixation, the plates were stained with 0.3% crystal violet in methanol at RT(Room temperature) for 15 min. After staining, the plate was washed by flowing water. Cell colonies were visualized using a microscope and quantified by counting or dissolving crystal violet in Sorenson's Buffer then analyzed by OD 570 nm.

## Human phospho-RTK antibody array

Proteome Profiler Human phosphor-RTK array kit (ARY001B; R&D Systems, Minneapolis, MN) was used to detect potential activation of RTK signals by ectopic expressed SULF1 stable transfectants. All procedures were conducted according to the manufacturer's instruction. Shortly, capture antibodies for specific RTKs were spotted in duplicate onto nitrocellulose membranes, provided by the kit. Suggested cell lysates were incubated with the array membrane at 4°C overnight. To wash the unbound material, protein with phosphorylated tyrosine residues from the extracted cell lysates were bound to the capture antibodies of RTK and detected by a pan-phospho-tyrosine antibody conjugated with HRP. Finally, the binding signal was determined by chemiluminescent reagent and exposed to x-ray film.

## Establishment of stable cell lines

Vector and SULF1 of JJ012 stable cell lines were established by pcDNA3.1(-) SULF1 transient transfection by jetPRIME (PO-114–15, Polyplus); Vector, WT SULF1, enzymatical inactive SULF1 (CA mutant), and ectopic expressed Luc stable cell lines were established by pcDNA3.1(-) SULF1, pcDNA3.1(-)

SULF1C 87 A/C88A and pcDNA3.1(-) Luc transient transfection by TransITR-2020 (MS-MIR5400, Mirus).

According to the type of vector, specific antibiotics were selected to generate the stable cells. G418 (1000 µg/ml) were used to selected JJ012 stable cells; G418 (200 µg/ml) for CH2879 stable cells. All stable cells were selected from a single clone. G418 sulfate (Cat. No. Ant-gn-1, Invivogen) was purchased from InvivoGen Company.

## Flow cytometric analysis

For the analysis of sulfated glucosamine containing HSPG on cell membrane, $3 \times 10^6$ of CH2879 and JJ012 stable cell lines were collected in 3% BSA and stained with 10E4 primary antibody (1:200 for 30 min; GTX20073), which recognizes native HS containing the N-sulfated glucosamine moiety. After primary antibody staining, cells were further stained with Mouse IgM FITC (1:200) for 30 min at 4°C in the dark. The samples and data were analyzed by FACSVerse (BD) and its software BD FACSuite v1.0.6.

## Immunohistochemical staining and scoring of human bone sarcoma

Human bone sarcoma tissue array (B0481, B0481a) was purchased from US Biomax Inc, and normal primary cartilage tissue array were collected from Dr. Teng-Le Huang (CMUH IRB No.102-REC1-047). Paraffin blocks were deparaffinized and hydrated through an ethanol series. Antigen retrieval were conducted by the pressure cooker contains a retrieval solution (Tris-EDTA pH 9.0), and retrieval enzyme (ab970, abcam). Next, slides were treated with hydrogen peroxide and blocked by blocking buffer and followed by incubated with corresponding primary antibodies (pcMET 1:100, #3077 Cell Sinaling, cMET 1:100 #8198 Cell Sinaling, SULF1 1:100 ab32763, Abcam) overnight at 4°C. The staining procedures were referred to manufactory's instructions (Leica, RS-I-515–1). The Digital images of IHC-stained slides were obtained at 40× magnification ($0.0625 \mu m^2$ per raw image pixel) using a whole slide scanner (Leica, Aperio CS2). Protein expression was ranked according to Histoscore (H-score) method. H-score was examined by a semi-quantitative evaluation of both the intensity of staining and percentage of positive cells (quantitative by Leica, image scope software, use Color Dconvolution v9 algorithm). The range of scores was from 0 to 250.

## Animal studies

All animal experiments were conducted according to animal welfare guidelines approved by China Medical University's institutional Animal care and Use committee (IACUC). The approval protocols were also included the ethics of animal experiments referring to 3 R (reduction, replacement, and refinement) principle (104–185 N, CMUIACUC-2018–329, and CMUIACUC-2020–120). For subcutaneous tumor model, 6-week-old female Balb/c nu/nu mice were inoculated $6 \times 10^6$ vector SULF1 CH2879 stable cells. For tumorigenicity of vector, WT SULF1, and CA mutant SULF1 stable cell lines, 6-week-old female Balb/c nu/nu mice were subcutaneously inoculated $6 \times 10^6$ cells contain matrix gel in final 4 mg/ml concentration. Tumors were measured by vernier calipers every 4 days, and the tumor volume was calculated using the following formula: $(V) = \pi/6 \times$ larger diameter $\times$ smaller diameter$^2$. The mice were sacrificed after 45 days. In orthotopic mice model, 6-week-old female Balb/c nu/nu mice were inoculated $5 \times 10^5$ CH2879 cells ectopic expressed with luciferase. Cells were resuspended in 20 µl PBS and injected into mouse tibial. After 1-week tumor was formed and followed by treated with crizotinib via oral gavage 5 days per week. The tumor volume was measure by the IVIS system, and mice were weighed once a week. The mice survival rate was following the signs of either moribundity or mortality.

## Statistical analysis

Each experiment was repeated three times. Unless otherwise noted, data are presented as mean $\pm$ SD. Statistical significance was determined by the Student $t$ test (two-tailed and unpaired). The statistical analysis for each plot is described in the figure legends. $p < 0.05$ was considered statistically significant.

## Acknowledgements

We would like to acknowledge Dr. JA Block (Rush University Medical Center, Chicago, IL, USA), who provided us with the JJ012 cell line and Professor A Llombart Bosch (University of Valencia, Spain) for the CH2879 cell line. This study was supported by the following: Ministry of Science and Technology (MOST: 107–2320-B-039–003; To Y-G C; 108–2320-B-039–003; To Y-H .; 109–2320-B-039–012; To Y-H. C); Drug Development Center, China Medical University from the Featured Areas Research Center program within the framework of the Higher Education Sprout Project by the Ministry of Education (MOE) in Taiwan (To Y-H C, and To M-C H); China Medical University (CMU 107-TU-07 to Y-H C; 107 S-36 to Y-H C; 109-MF-27 to Y-H. C). MOST 110–2639-B-039–001 -ASP (to M-C. H).

## Additional information

### Funding

| Funder | Grant reference number | Author |
| --- | --- | --- |
| Ministry of Science and Technology, Taiwan | 107-2320-B-039-003 | Ya-Huey Chen |
| Ministry of Science and Technology, Taiwan | 108-2320-B-039-003 | Ya-Huey Chen |
| Ministry of Science and Technology, Taiwan | 109-2320-B-039-012 | Ya-Huey Chen |
| Ministry of Education | Featured Areas Research Center program within the framework of the Higher Education Sprout Project | Ya-Huey Chen Mien-Chie Hung |
| China Medical University, Taiwan | 107-TU-07 | Ya-Huey Chen |
| China Medical University | 107-S-36 | Ya-Huey Chen |
| Ministry of Science and Technology, Taiwan | 110-2639-B-039-001 -ASP | Mien-Chie Hung |
| China Medical University | 109-MF-27 | Ya-Huey Chen |

The funders had no role in study design, data collection and interpretation, or the decision to submit the work for publication.

### Author contributions

Zong-Shin Lin, Data curation, Software, Formal analysis, Validation, Investigation, Visualization, Methodology; Chiao-Chen Chung, Yu-Chia Liu, Yung-Yi Liang, Software, Formal analysis, Validation, Investigation, Visualization, Methodology; Chu-Han Chang, Hui-Chia Liu, Validation, Investigation; Teng-Le Huang, Resources, Methodology; Tsung-Ming Chen, Conceptualization, Resources, Investigation; Che-Hsin Lee, Chih-Hsin Tang, Conceptualization, Investigation; Mien-Chie Hung, Conceptualization, Supervision, Writing - review and editing; Ya-Huey Chen, Conceptualization, Resources, Data curation, Supervision, Investigation, Methodology, Writing - original draft, Project administration, Writing - review and editing

### Author ORCIDs

Zong-Shin Lin http://orcid.org/0000-0003-3365-1240
Mien-Chie Hung http://orcid.org/0000-0003-4317-4740
Ya-Huey Chen http://orcid.org/0000-0002-5757-8634

### Ethics

Human subjects: Specific ethical approval and guidelines were obtained and followed by the Research Ethics Committee of China Medical University Hospital. Based on the agreement of the informed consent form, the cartilage tissues were obtained from the patients with surgery of total knee

replacement. The results of studies would be allowed to publish with patient information confidentiality, also included in the consent form. IRB ID: CMUH IRB No.102-REC1-047.

The animal experiment of this study was performed in strict accordance with the recommendations in the Guide for the Care and Use of Laboratory Animals of the Council of Agriculture, Executive Yuan, TW. All of the animals were handled following the protocol of the laboratory animal center of China Medical University. All surgery was performed under isoflurane anesthesia, and every effort was made to minimize suffering. The approval protocols also included the ethics of animal experiments referring to the 3R (reduction, replacement, and refinement) principle (104-185-N, CMUIACUC-2018-329, and CMUIACUC-2020-120).

### Decision letter and Author response
Decision letter https://doi.org/10.7554/eLife.79432.sa1
Author response https://doi.org/10.7554/eLife.79432.sa2

---

## Additional files

### Supplementary files
• MDAR checklist

### Data availability
All data generated or analyzed during this study are included in the manuscript and supporting file.

The following dataset was generated:

| Author(s) | Year | Dataset title | Dataset URL | Database and Identifier |
|---|---|---|---|---|
| Lin Z, Chung C, Liu Y, Chang C, Liu H, Liang Y, Huang T, Chen T, Lee C, Tang C, Hung M, Chen Y | 2022 | EZH2/ hSULF1 axis mediates receptor tyrosine kinase signaling to shape cartilage tumor progression | https://doi.org/10.5061/dryad.jwstqjqdp | Dryad Digital Repository, 10.5061/dryad.jwstqjqdp |

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

# Appendix 1

**Appendix 1—key resources table**

| Reagent type (species) or resource | Designation | Source or reference | Identifiers | Additional information |
|---|---|---|---|---|
| Cell line (*Homo sapiens*, chondrosarcoma) | JJ012 | Dr. JA Block (Rush University Medical Center, Chicago, IL, USA) | RRID: CVCL_D605 | |
| Cell line (*Homo sapiens*, chondrosarcoma) | CH2879 | Professor A Llombart Bosch (University of Valencia, Spain) | RRID: CVCL_9921 | |
| Cell line (*Homo sapiens*, osteosarcoma) | U2OS | From Dr. Jer-Yuh Liu | RRID:CVCL_0042 | |
| Cell line (*Homo sapiens*, osteosarcoma) | MG63 | From Dr, Shian-Ying Sung | RRID:CVCL_0426 | |
| Cell line | HEK293T | Academia Sinica, Taiwan | RRID: CVCL_0063 | For lentiviral production |
| Cell line | NP-Chon | From bone surgeon Dr.Teng-Le Huang | Detail information was in the supplement file | Isolated from clinical tissue sample |
| Biological sample | Cartilage tissue | From bone surgeon Dr.Teng-Le Huang | Detail information was in the supplement file | Isolated from clinical tissue sample |
| Strain and strain background | DH5α | This paper | | Competent cell for construction |
| Strain and strain background | Lentivirus | Academia Sinica, Taiwan | | For shRNA transfection |
| Strain and strain background | NU/NU (Crl:NU-Foxn1nu)/ female | BioLASCO | | For animal study |
| Transfected construct (human) | shEZH2#3 | Academia Sinica, Taiwan | TRCN0000040076 | Lentiviral construct to transfect and express the shRNA. |
| Transfected construct (human) | shEZH2#4 | Academia Sinica, Taiwan | TRCN0000010475 | Lentiviral construct to transfect and express the shRNA. |
| Transfected construct (human) | shSULF1#7 | Academia Sinica, Taiwan | TRCN0000373658 | Lentiviral construct to transfect and express the shRNA. |
| Transfected construct (human) | shSULF1#8 | Academia Sinica, Taiwan | TRCN0000373588 | Lentiviral construct to transfect and express the shRNA. |
| Transfected construct (human) | pcDNA3.1(-)-Luc | This paper | | Transfected construct for luciferase expression (human) |
| Transfected construct (human) | pcDNA3.1(-)-SULF1 WT | This paper | | Transfected construct for WT SULF1 expression (human) |
| Transfected construct (human) | pcDNA3.1(-)-SULF1 CA | This paper | | Transfected construct for enzyme inactivated SULF1 expression (human) |
| Antibody | ß-Actin | Santa Cruz Biotechnology (mouse monoclonal) | sc-47778 RRID:AB_626632 | WB 1:10000 |
| Antibody | pAKT | Cell Signaling Technology (rabbit polyclonal) | sc-7985-R RRID:AB_2861344 | WB 1:1000 |

*Appendix 1 Continued on next page*

*Appendix 1 Continued*

| Reagent type (species) or resource | Designation | Source or reference | Identifiers | Additional information |
|---|---|---|---|---|
| Antibody | AKT | Cell Signaling Technology (rabbit polyclonal) | #9272 RRID:AB_329827 | WB 1:1000 |
| Antibody | pCMET | Cell Signaling Technology (rabbit polyclonal) | #3077 RRID:AB_2143884 | IHC: 1:100 WB 1:1000 |
| Antibody | pCMET | ABclonal (rabbit polyclonal) | AP0533 RRID:AB_2771334 | WB 1:1000 |
| Antibody | CMET | Cell Signaling Technology (rabbit polyclonal) | #8198 RRID:AB_10858224 | WB 1:1000 |
| Antibody | CD44v6 | R&D systems Technology (mouse monoclonal) | BBA13 RRID:AB_356935 | WB 1:1000 |
| Antibody | pERK 1/2 | Genetex Biotechnology (rabbit polyclonal) Cell Signaling Technology (rabbit polyclonal) | GTX59568 RRID:AB_10731702 #9101 RRID:AB_331563 | WB 1:1000 |
| Antibody | ERK 1/2 | Cell Signaling Technology (rabbit monoclonal) | #4695 RRID:AB_390779 | WB 1:1000 |
| Antibody | H3K27me3 | Cell Signaling Technology (rabbit polyclonal) Abcam (mouse monoclonal) | #9733 RRID:AB_2616029 ab6002 RRID:AB_305237 | WB 1:1000 |
| Antibody | Histone 3 | Genetex Biotechnology (rabbit polyclonal) Cell Signaling Technology (rabbit polyclonal) | GTX122148 RRID:AB_10633308 #9715 RRID:AB_331563 | WB 1:1000 |
| Antibody | pP38 | Cell Signaling Technology (rabbit polyclonal) | #9211 RRID:AB_331641 | WB 1:1000 |
| Antibody | P38 | Santa Cruz Biotechnology (rabbit polyclonal) | sc-7149 RRID:AB_653716 | WB 1:1000 |
| Antibody | SULF1 | Abcam (rabbit polyclonal) | ab327 RRID:AB_882749 63 | IHC: 1:100 WB 1:1000 |
| Antibody | EZH2 | Cell Signaling Technology (rabbit polyclonal) | #3147 RRID:AB_10694383, #5246 RRID:AB_10694683 | IHC: 1:100 WB 1:1000 |
| Antibody | 10E4 | Amsbio (mouse monoclonal) | #370255 S RRID:AB_10891554 | WB 1:1000 |
| Antibody | Secondary antibody-HRP conjugated | Bioss antibodies (Goat polyclonal) | BS-0368G-HRP RRID:AB_10890902 | WB 1:5000 |
| Antibody | Secondary antibody-HRP conjugated | Mouse IgG antibody (HRP) (Rabbit polyclonal) | GTX213112-01 RRID:AB_10617557 | 1:5000 |

*Appendix 1 Continued on next page*

*Appendix 1 Continued*

| Reagent type (species) or resource | Designation | Source or reference | Identifiers | Additional information |
|---|---|---|---|---|
| Antibody | Secondary antibody-HRP conjugated | Rabbit IgG antibody (HRP) (Goat polyclonal) | GTX213110-01 RRID:AB_10618573 | 1:4000 |
| Sequence-based reagent | EZH2-q-F | This paper | qPCR-Primers | 5'-CAGTTCGTGCCCTTGTG TGA-3' |
| Sequence-based reagent | EZH2-q-R | This paper | qPCR-Primers | 5'-GCACTGCTTGGTGTTGC ACT-3' |
| Sequence-based reagent | SULF1-q-F | This paper | qPCR-Primers | 5'CAAGGAGGCTGCTCAGGA AG3' |
| Sequence-based reagent | SULF1-q-R | This paper | qPCR-Primers | 5'CATGCGTGAAGCAAGTGA GG3' |
| Sequence-based reagent | q-ChIP-SULF1-F-P | This paper | qPCR-Primers | 5'CGCATGCGGAATGACAAC AG3' |
| Sequence-based reagent | q-ChIP-SULF1-R-P | This paper | qPCR-Primers | 5'CTCAGTTCAAATCCCGCC TC3' |
| Chemical compound and drug | Capmatinib | Selleckchem | INCB28060 | For pCMET inhibition |
| Chemical compound and drug | Crizotinib | Sigma | PZ0191 | For pCMET inhibition |
| Chemical compound and drug | EPZ-6438 | MCE | HY-13803 | For EZH2 enzyme activity inhibition |
| Chemical compound and drug | GSK343 | Sigma | SML0766 | For EZH2 enzyme activity inhibition |
| Chemical compound and drug | G418 (Geneticin) | InvivoGen | Anti-gn-1 | For stable cell line selection |
| Chemical compound and drug | Tivantinib | Selleckchem | S2753 | For pCMET inhibition |
| Commercial assay and kit | Human RTK array | R&D Systems | ARY001B | For human RTK receptor detection |
| Commercial assay and kit | MTS | Promega | RG3580 | For cell proliferation detection |
| Commercial assay and kit | MTT | InvivoGen | M6494 | For cell proliferation detection |
| Commercial assay and kit | SYBR green | Roche | KK4600 | For mRNA detection |
| Commercial assay and kit | Antigen retrieval solution | Abcam | ab970 | For antigen retrieval |
| Commercial assay and kit | Lipofectamine | Invitrogen | 11668019 | |
| Commercial assay and kit | TransITR-2020 | Mirus | MS-MIR5400 | |
| Commercial assay and kit | Jet PRIME | Polyplus | PO-114–15 | |

*Appendix 1 Continued on next page*

*Appendix 1 Continued*

| Reagent type (species) or resource | Designation | Source or reference | Identifiers | Additional information |
| --- | --- | --- | --- | --- |
| Software | Image Lab | Bio-Rad Laboratories | RRID:SCR_014210 | For WB image analysis |
| Software | BD FACSuite v1.0.6 | BD bioscience | | For FACS analysis |
| Software | IVIS Spectrum In Vivo Imaging System | PerkinElmer | | For in vivo tumour analysis |
| Software and algorithm | PRISM | GraphPad Software | RRID:SCR_002798 | For survival analysis and bargraph |
| Software and algorithm | Image scope | Leica | | For IHC H score quantification |
| Other | Bone cancer tissue array | US Biomax | B0481 | Chondrosarcoma IHC staining |
| Other | Bone cancer tissue array | US Biomax | B0481a | Chondrosarcoma IHC staining |
| Other | Normal cartilage tissue array | From bone surgeon Dr.Teng-Le Huang | | Primary cartilage tissue IHC staining |

