## [Editor Report]

This fundamental work by Hung et al. substantially advances our understanding of the biology of chondrosarcoma, a primary cancer of cartilaginous tissue that can progress to highly aggressive, metastatic, and treatment-refractory tumor. The authors provided compelling to exceptional evidence to support the conclusion that EZH2/ hSULF1 axis mediates receptor tyrosine kinase signaling to shape cartilage tumor progression. This work will be of broad interest to cancer biologists and oncologists.

---

## [Decision Letter]

**Decision letter after peer review:**

Thank you for submitting your article "EZH2/ hSULF1 axis mediates receptor tyrosine kinase signaling to shape cartilage tumor progression" for consideration by *eLife*. Your article has been reviewed by 2 peer reviewers, and the evaluation has been overseen by a Reviewing Editor and Wafik El-Deiry as the Senior Editor. The following individual involved in the review of your submission has agreed to reveal their identity: Mickey C-T. Hu (Reviewer #2).

Essential revisions:

1) Please address points 1-3 raised by Reviewer #1 through new experimentation or clarification.

*Reviewer #1 (Recommendations for the authors):*

Below are the comments that can help further strengthen the study.

1. Is EZH2 enzymatic activity required for hSULF1 repression?

2. Does the loss of EZH2, which induces hSULF1 expression, also impair c-Met phosphorylation as hSULF1 overexpression?

3. It will further strengthen the concept that SULF1 acts through its phosphatase activity to restrict tumorigenesis by including the SULF1 enzymatic dead mutant in the tumorigenesis assay presented in Figure 3F.

4. It will be beneficial for this study to compare the efficacy of EZH2 inhibitor and c-Met inhibitor in suppressing tumor growth in an in vivo tumor model in Figure 7C.

*Reviewer #2 (Recommendations for the authors):*

Lin et al. demonstrate a novel and crucial discovery concerning the role of the EZH2/SULF1/cMET signaling pathway in controlling the pathological progression of chondrosarcoma, which is a malignant cartilaginous tumor and a common type of primary bone cancer. Impressively, this manuscript validates the therapeutic significance of this pathway in chondrosarcoma cells in vitro and in vivo (in mouse tumor models) using clinically used specific EZH2 and cMET inhibitory drugs that significantly contribute to the therapeutic potential for the future development of effective targeted therapy for chondrosarcoma. Moreover, this manuscript illustrates the pathological significance of the SULF1/cMET pathway in chondrosarcoma; thus, after meticulously reading this manuscript, I believe this publication has a high impact in the field and strongly recommend the Editor accept this manuscript for publication with my highest enthusiasm.

---

## [Author Response]

Reviewer #1 (Recommendations for the authors):Below are the comments that can help further strengthen the study.1. Is EZH2 enzymatic activity required for hSULF1 repression?

To address this point, we performed the new experiment (new Figure 4D in the revised version, attached below) and the results support “EZH2 enzymatic activity required for hSULF1 repression”. In brief, we explored hSULF1 expression in chondrosarcoma cell line treated with EPZ-6438, an EZH2 inhibitor, to block EZH2 enzymatic activity. The expression of hSULF1 was increased in a dose dependent manner in chondrosarcoma cell line in present of EPZ6438 (new Figure 4D). Also, the phosphorylation of MET was repressed while treated with EZH2 inhibitor of chondrosarcoma cell line. In addition, In the experiment shown in Figure 2E and F, we assessed the expression of hSULF1 in chondrosarcoma cell lines with *EZH2* specific shRNA, and hSULF1 expression was augmented while *EZH2* was knockdown. Together, the data showed that EZH2 enzyme activity is required for the expression of hSULF1 in chondrosarcoma cell lines.

2. Does the loss of EZH2, which induces hSULF1 expression, also impair c-Met phosphorylation as hSULF1 overexpression?

We thank the reviewer for the comments. Following the original data shown in Figure 2E and F, we repeated the experiment and included analysis of phosphorylation of cMET, the results demonstrated “the loss of EZH2, which induces hSULF1 expression, also impairs c-Met phosphorylation as hSULF1 overexpression (New in Figure 4E in revised version).

3. It will further strengthen the concept that SULF1 acts through its phosphatase activity to restrict tumorigenesis by including the SULF1 enzymatic dead mutant in the tumorigenesis assay presented in Figure 3F.

We appreciate the reviewer for the comments and performed the suggested experiments (New Figure 4I in revised version) which allowed us to identify an unexpected phosphorylation of Y1003 of cMET in tumors that may be worthy of further pursuing in the near future.

Chondrosarcoma stable cell lines with vector control, wild type SULF1 (WT), and enzymatic inactive mutant SULF1 (CA) were subcutaneously injected into the nude mice. Tumor volume was measured twice per week. The results indicated that WT SULF1, associated with full sulfatase activity inhibited tumor development, consistent to what was observed in vitro culture. But the CA mutant, which lost sulfatase activity, thought to recover tumorigenicity, still associated with partial tumor suppression activity as it could not completely recover tumor volume similar to that of vector control (New Figure 4I in revised version).

To further elucidate the possible underling mechanism for why the CA mutant which lost sulfatase activity still associated with partial tumor suppression activity. We examined the different phosphorylation site of cMET and found that CA mutant retained the similar phosphorylated profile of Y1234/1235 of cMET, a catalytic activation site, in cell lysates and tumors (New Figure 4 – supplement 1A in revised version). However unexpectedly, the gain of phosphorylation of Y1003 of cMET was markedly detected in tumors, but not in the in vitro cell culture (New New Figure 4 – supplement 1B in revised version). The phosphorylation of Y1003 of cMET has been implicated to associate tumor suppressive activity (Organ and Tsao, 2011). It is not yet clear how the tumor microenvironment might cause the unexpected results of CA mutant in the animal study (New Figure 4I) and it is certainly interesting to pursue further how tumor microenvironment might affect p-Y1003 of cMET in the future.

4. It will be beneficial for this study to compare the efficacy of EZH2 inhibitor and c-Met inhibitor in suppressing tumor growth in an in vivo tumor model in Figure 7C.

We thanks the reviewer comments. The treatment of EZH2 inhibitor in chondrosarcoma mice model was important. However, we had been tested several EZH2 inhibitors including GSK343 and EPZ-6438(Tazmetostat) in vitro (SFigure 3) and indicated that range of IC50 was 6 -14μM, much higher than cMET inhibitor (Crizotinib; IC50 = ~3μM) (SFigure 4). Since the IC50 of EZH2 inhibitors are much higher than that of c-Met inhibitors, we tested in vivo tumor assay using c-Met inhibitor ( Figure 7). In the future, if more advanced EZH2 inhibitors are developed further, it can be tested in the future.

Reference:

Organ, S. L., and Tsao, M. S. (2011). An overview of the c-MET signaling pathway. Ther Adv Med Oncol, 3(1 Suppl), S7-S19. doi:10.1177/1758834011422556